# Unlocking the Secrets of Cancer Stem Cells with γ-Secretase Inhibitors: A Novel Anticancer Strategy

**DOI:** 10.3390/molecules26040972

**Published:** 2021-02-12

**Authors:** Maryam Ghanbari-Movahed, Zahra Ghanbari-Movahed, Saeideh Momtaz, Kaitlyn L. Kilpatrick, Mohammad Hosein Farzaei, Anupam Bishayee

**Affiliations:** 1Medical Technology Research Center, Health Technology Institute, Kermanshah University of Medical Sciences, Kermanshah 6734667149, Iran; Maryam.gh.movahed@gmail.com; 2Department of Biology, Faculty of Science, University of Guilan, Rasht 4193833697, Iran; 3Medical Biology Research Center, Kermanshah University of Medical sciences, Kermanshah 6714415185, Iran; Zahra.gh.movahed@gmail.com; 4Medicinal Plants Research Center, Institute of Medicinal Plants, Academic Center for Education, Culture and Research, Tehran 1417614411, Iran; saeideh58_momtaz@yahoo.com; 5Toxicology and Diseases Group, The Institute of Pharmaceutical Sciences, Tehran University of Medical Sciences, Tehran 1417614411, Iran; 6Gastrointestinal Pharmacology Interest Group, Universal Scientific Education and Research Network, Tehran 1417614411, Iran; 7Lake Erie College of Osteopathic Medicine, Bradenton, FL 34211, USA; kilpatrickkaitlyn@gmail.com

**Keywords:** cancer stem cells, γ-secretase, γ-secretase inhibitors, Notch signaling, cancer treatment

## Abstract

The dysregulation of Notch signaling is associated with a wide variety of different human cancers. Notch signaling activation mostly relies on the activity of the γ-secretase enzyme that cleaves the Notch receptors and releases the active intracellular domain. It is well-documented that γ-secretase inhibitors (GSIs) block the Notch activity, mainly by inhibiting the oncogenic activity of this pathway. To date, several GSIs have been introduced clinically for the treatment of various diseases, such as Alzheimer’s disease and various cancers, and their impacts on Notch inhibition have been found to be promising. Therefore, GSIs are of great interest for cancer therapy. The objective of this review is to provide a systematic review of in vitro and in vivo studies for investigating the effect of GSIs on various cancer stem cells (CSCs), mainly by modulation of the Notch signaling pathway. Various scholarly electronic databases were searched and relevant studies published in the English language were collected up to February 2020. Herein, we conclude that GSIs can be potential candidates for CSC-targeting therapy. The outcome of our study also indicates that GSIs in combination with anticancer drugs have a greater inhibitory effect on CSCs.

## 1. Introduction

Despite the remarkable progress being made in cancer treatment, cancer is the leading cause of death worldwide. There is evidence that a rare subset of cancer cells are responsible for resistance to therapy and holding stemness functions/properties, which are known as cancer stem cells (CSCs). Therefore, these cell subpopulations induce tumor perpetuation, even after effective therapies, and result in aggression of the tumor. The CSC theory of cancer progression proposes that a tumor is a hierarchically organized tissue with CSCs at the top position in the hierarchy, which produces more differentiated cancer cells with a reduced or restricted potential of proliferation. In recent years, the CSC theory has been subjected to increasing attention and excitement, with researchers believing this theory would augment our knowledge about the molecular and cellular events within tumor progression, contributing to metastasis, recurrence, and resistance to therapy [1].

CSCs possess similar stemness properties to normal stem cells, including the differentiation, proliferation, and self-renewal capabilities that create heterogeneous cancer cell populations. Considering these similarities, normal stem cells and CSCs are usually characterized by the cell surface markers, such as CD20, CD90, CD133, CD44, and CD166 [2]. These markers are used to isolate CSCs by magnetic-activated cell sorting or fluorescence-activated cell sorting techniques [3,4]. Additionally, there are various functional assays for the isolation of CSCs, such as side population cells, tumorigenicity, immune selection by natural killer cells, the aldehyde dehydrogenases (ALDH) assay, and label-retaining methods [3]. Moreover, their tumor-forming capacity is characterized by their increased tumor-repopulating ability when transplanted into immunodeficient mouse models. Additionally, sphere formation assays, as in vitro assays, are used for the enrichment and identification of CSCs [5].

To date, numerous cellular signaling pathways and mediators have been identified, which are potentially able to mediate the carcinogenesis process. The Notch pathway is a highly conserved signaling pathway that plays a critical role in proliferation, development, maintenance of stem cell and multicellular organism homeostasis, differentiation, and specification of cell fate [6]. The Notch signaling pathway also contributes to connections between the self-renewal ability of CSCs and angiogenesis and is therefore gaining attention for targeting CSCs [7]. Notch signaling activation mostly relies on the activity of the γ-secretase enzyme that cleaves the Notch receptors and releases the active intracellular domain. Therefore, γ-secretase inhibitors (GSIs) are promising therapeutic targets for the suppression of Notch signaling. GSIs were the first class of compounds to reach clinical development in cancer research and can be divided into three classes, namely, sulfonamides, azepines, and peptide isosteres, of which sulfonamides and azepines are more commonly used [8]. Several studies have shown that these inhibitors possess anticancer, anti-CSC, antiangiogenesis, and antitumor growth activities and can cause apoptosis, especially in combination with targeted chemotherapeutic drugs. Therefore, the blockade of Notch signaling by GSIs might be a promising target for cancer therapy through the complete eradication of CSCs [9]. Despite the availability of GSIs, Notch-related treatments are presently prohibited by side effects, because of the requirement for the Notch pathway in most tissues [10]. Using nano-delivery systems for CSC-targeted therapy is one of the strategies for overcoming these challenges [11].

Although a few reports present an overview of GSIs, these publications evaluate the antitumor activities of GSIs in a limited number of cancers or review the pharmacological activities of GSIs without a particular emphasis on their anticancer effects [12,13,14]. Moreover, a comprehensive and critical systematic review on the effect of GSIs on CSC elimination within different cancers has not been conducted in the past. Therefore, the present systematic review was conducted to critically evaluate the results obtained from studies on the effect of GSIs on the elimination of a wide variety of CSCs.

## 2. The Role of the Notch Signaling Pathway in Cancer

Notch signaling starts with Notch receptors binding to ligands of the Delta-like ligand (Dll) and Jagged families, leading to successive cleavages, first in the extracellular domain, and then in the transmembrane domain, which release the intracellular domain of Notch (ICN) and allow its translocation to the nucleus. In the nucleus, ICN enhances the expression of downstream targets such as the Hairy/Enhancer of Split (HES) gene via binding the transcription factor CSL (CBF1, suppressor of hairless and LAG1), and mastermind-like polypeptides (MAML) [15].

One of the primary connections between cancer and Notch signaling was found in 1991 in human T-cell acute lymphoblastic leukemia [16]. In B-cell malignancies, such as chronic lymphocytic leukemia, Notch-1 mutations were associated with an enhanced progression of disease and chemotherapy resistance [17]. In addition to the dysfunction of Notch receptors in leukemia, the ligand Jagged-2 is considerably overexpressed in multiple myeloma (Figure 1) [18].

The oncogenic reach of the Notch pathway is partially due to its crosstalk with other signaling pathways; for example, Hedgehog, Janus kinase (JAK)/signal transducers and activators of transcription (STAT), receptor tyrosine kinases (RTKs), Wnt, and transforming growth factor-β (TGF-β) decapentaplegic pathways. In addition to TGF-β and Wnt, vascular endothelial growth factor (VEGF), phosphatidylinositol 3-kinase (PI3K)/Akt, Ras, nuclear factor κ-light-chain-enhancer of activated B cells (NF-κB), mammalian target of rapamycin (mTOR), hypoxia-inducible factor (HIF), and interleukin-6 (IL-6) are pertinent to Notch crosstalk [19]. It has been indicated that NF-κB signaling regulates the Notch pathway and is regulated by the Notch pathway. For instance, the intracellular domains of Notch3 and Notch1 have been observed to stimulate NF-κB signaling components, such as the inhibitor of κB kinase (IKK)) [20]. Indeed, based on evidence of Notch signaling’s role in cancer progression and development, this pathway has become a main target for new therapeutic treatments in this field [21].

## 3. The Role of GSIs in Notch Signaling Pathway Inhibition

Depending on the chemical structure and binding sites, GSIs can be divided into two large classes, namely peptide inhibitors and non-peptide inhibitors. Peptide inhibitors are divided into five sub-classes, namely peptide aldehyde derivatives, difluoroketone derivatives, hydroxyethylene dipeptide isostere derivatives, α-helical peptide derivatives, and dipeptide analogues. Non-peptide inhibitors represent two sub-classes, namely benzodiazepines derivatives and sulfonamides derivatives [22].

The peptide inhibitors interact well with two aspartates present at the active site, but are not susceptible to cleavage via the action of protease. For example, difluoroketone peptidomimetic inhibitors, such as difluoroketone-167 (DFK-167), can directly bind to the active site. The non-peptide inhibitors, such as LY-411,575, *N*-[*N*-(3,5-difluorophenacetyl)-l-alanyl]-S-phenylglycine t-butyl ester (DAPT), and LY-450,139, bind to docking sites and consist of non-competitive γ-secretase inhibitors. These inhibitors suppress the S3 cleavage of the Notch receptors to block the Notch activity [23].

MK-0752 (cis-3-[4-[(4-chlorophenyl)sulfonyl]-4-(2,5-difluorophenyl)cyclohexyl] propanoic acid), which is a potent GSI with an adequate oral bioavailability, blocks NICD cleavage and its subsequent nuclear translocation [24]. Another GSI, known as RO4929097, which is a dipeptide analogue, has been designed to be used as a chiral building block for the preparation of malonamide derivatives which can act as a GSI [23]. One study showed that RO4929097 could inhibit tumor growth and has on-target pharmacodynamic effects in preclinical models of blood cancer [25]. L-685,458, which belongs to the hydroxyethylamines, contains a hydroxyethylene dipeptide isostere and can function as a transition state analogue mimic of an aspartyl protease. PF-03084014, which belongs to dipeptide analogues, is a selective, noncompetitive, and reversible GSI [23]. One study indicated that PF-03084014 reduced mammosphere formation in vitro, and showed antimetastatic and antitumor activity in various breast xenograft models [26]. It was shown that DAPT in combination with taxanes induced cycle arrest in colorectal cancer (CRC) cells [27]. Another study used GSIs such as dibenzazepine (DBZ), L-685,458, and DAPT to examine their impact on the survival or growth of a CRC cell line. Surprisingly, this study could not elicit major CRC cell growth suppression using these GSIs. Nevertheless, treatment with these compounds significantly decreased the Val1744-NICD fragment abundance (a Notch fragment that is greatly detectable in a subset of CRC cell lines) after a few hours. DBZ appeared more effective and persistent. This result showed that GSIs in combination with chemotherapy were more effective than GSIs alone. In the same study, a combination of GSIs and platinum-based therapy, in particular cisplatin, was capable of enhancing the cell death in CRC [28]. Another study revealed that the inhibition of DLL4- and DLL1-mediated Notch signaling led to a loss of intestinal goblet cells, but inducible Jagged-1 deletion had no obvious phenotype [29]. Moreover, it was reported that MRK-003 treatment reduced the tumor onset and tumor burden in BALB/c-neuT female mice bearing ERBB2-positive breast cancer cells [30]. The chemical structures of several GSIs are presented in Figure 2. Despite their different chemical structures, all GSIs that impair Notch signaling act through its suppression [8,31].

## 4. Methodology for the Literature Search and Study Selection

The current systematic review was conducted following the Preferred Reporting Items for Systematic Reviews and Meta-Analysis (PRISMA) guidelines [32]. The objective of this review is to provide a systematic review of in vitro and in vivo studies for investigating the impact of GSIs on cancer stem cells. Various electronic scholarly databases, including Scopus, PubMed, Science Direct, Google Scholar, and Web of Science, were searched and relevant studies in English language were collected up to February 2020. The search syntax included “cancer stem cell” OR “tumor stem cell” OR “initiating tumor cell” OR “neoplastic stem cell” OR “colony forming unit” AND “gamma-secretase inhibitor” OR “γ-secretase inhibitor”. The primary search was conducted by two researchers separately, and unrelated studies were excluded based upon their titles and abstracts. Review articles, meta-analyses, books, book chapters, conference abstracts, case reports, clinical trials, and non-English articles were also excluded. An overview of the literature search and selection process is presented in Figure 3.

## 5. Anticancer Activities of GSIs against CSCs

Among 118 eligible articles, 64 and 4 studies were performed using in vitro cancer cell lines and in vivo animal models, respectively, and in 50 studies, both in vitro and in vivo models were used. Considering the major aspects of the total included studies and based upon the location of the cancer, the results are presented in the next sections (Table 1 and Table 2).

### 5.1. Adenoid Cystic Carcinoma

A study by Panaccione et al. [33] investigated the anticancer mechanism of GSIs in adenoid cystic carcinoma (ACC) cell lines under both in vitro and in vivo conditions. The in vitro study was performed by treating Accx11 cells with 1–10 μM DAPT with or without radiation for 24–96 h. For the in vivo assay, mice were treated with 50 mg/kg DAPT for 35 days. The treatment blocked S-phase kinase-associated protein 2 and the Notch 1 intracellular domain (N1-ICD), suppressed the growth of ACC in vivo, reduced CD133^+^ cells and sensitized them to radiation, and led to the induction of cyclin-dependent kinase inhibitor 1B (p27^Kip1^). Therefore, the combination of radiation and GSI exerted a greater effect on the elimination of CD133^+^ cells compared with either agent alone.

### 5.2. Blood Cancer

In an experiment, different lymphoma cell lines were exposed to 0.1–100 μM L685458 for 24 h and subjected to a colony-forming unit (CFU) assay. L-685458 potently blocked the CFU and reduced the lymphoma stem cell population [34].

In another study, Ikram et al. (2018) used a three-dimensional (3D) cell culture to assess the antitumor effects of DAPT. The pretreatment of lymphoma cells with DAPT (5 μM) with or without NSC23766 (Rac-specific small-molecule inhibitor) for 24 h showed a significant decrease of the lymphoma stem cell population and an increased sensitivity to doxorubicin [35].

The administration of varying concentrations of DAPT (8–16 μM) for 14 days decreased the colony number in leukemic stem cells, leading to a decline in the size of large colonies by suppressing their proliferation in a concentration-dependent way [36]. Furthermore, another study showed that treatment with MRK-003 (150 mg/kg) eliminated leukemia-initiating cells in a Tal1/Lmo2 mouse model of T-cell acute lymphoblastic leukemia (T-ALL) [37].

In a T-ALL cell line with NOTCH1 mutations (DND-41) and AML cell line (NB4), GSI-XXI (compound E, 10 μM) treatment, alone or in combination with cyclopamine or quercetin for 1–7 days, suppressed the colony formation ability of these cells. The addition of cyclopamine or quercetin to compound E enhanced the inhibitory effect on DND-41 cell line growth and blocked the activation of NOTCH1 in NB4 cell lines [38].

In another study, AA and HEL erythroid leukemia cells were cultured with three GSIs (GSI-IX, 20 μM; GSI-XII, 5 μM; and GSI-XXI, 10 μM) for 1–7 days. The study claimed that treatment with GSI induced the differentiation of morphologic erythroid and enhanced the production of hemoglobin. It also showed that treatment with GSIs inhibited the colony formation ability and short-term cell growth, while GSI-XXI treatment enhanced the AA cell line growth [39].

In multiple myeloma cancer stem cells (MM-CSCs), the effect of bruceantin (a quassinoid isolated from *Brucea* species) was evaluated in the presence of GSI. Bruceantin effectively controlled the MM-CSCs’ viability, migration, proliferation, and angiogenesis. MM-CSC pretreatment with the GSI (RO4929097, 10 μM) and increasing doses of bruceantin for 1 day inhibited the proliferation of these cells [40].

### 5.3. Brain Cancer

In brain cancer cell lines, it was established that the suppression of Notch signaling with DAPT inhibited hypoxia-induced GSC expansion [41]; abolished the effects of STC1 on N1-ICD production, SOX2 expression, and the sphere-forming capacity [42]; reduced the CSC of CD133^+^ and inhibited the proliferation of SHG-44 cells [43]; suppressed the transition from CD1331/CD1442 to double-positive (DP) [44]; inhibited cell growth and reduced the sphere formation capacity in glioblastoma neurosphere cultures [45]; and downregulated HIF-1α and hes1, reduced the number of nestin^+^ cells, increased the number of β-III-tubulin^+^ cells, and enhanced MKI67 and neuronal differentiation [46]. However, one study showed that DAPT treatment reduced brain CSCs, but had no survival benefit for mice injected with DAPT-treated GBM neurosphere cells [47].

DAPT treatment in combination with radiation [48], gleevec and amph1D peptide [49], D341Med with HBMEC [50], and imatinib [51], resulted in an increase of apoptosis and radio-sensitivity in ihBTC2 cells [48]; the induction of neurosphere dispersion that resulted in cell death [49]; the downregulation of Bmi-1, CDK6, c-Myc, and CCND1 expression in D341Med, and a reduction in the tumor size and volume [50]; and the effective growth inhibition of GBM cells [51].

The administration of DAPT and INCB3619 downregulated the expression of HES1 and HEY1 Notch target genes in both 0822 and 0308 cell lines. In the 0308 cell line, INCB3619 and DAPT also downregulated the expression of YKL-40/CHI3L1, while the survival was prolonged in mice [52].

In four different studies, DAPT, L685,458, BMS-708163, and RO4929097 treatment led to an increase of the ASCL1 levels in ASCL1^hi^ GSCs and a decrease in sphere-forming cells (SFCs) [53]; inhibited glioma tumor-initiating cell growth in a concentration-dependent manner, suppressed tumor growth, and prolonged the survival rate in vivo [54]; increased radiation-induced apoptosis and decreased the clonogenic survival of GSCs [55]; and decreased the number of CSCs by reducing proliferation and increasing cell death that was associated with decreased levels of STAT3 and Akt phosphorylation and resulted in the inhibition of tumor growth and enhancement of the survival rate [56].

Upon the usage of different concentrations of GSI-18 in vitro and in vivo, two studies reported a reduction in Hes1 protein and mRNA levels in DAOY cells, the suppression of clonogenicity, and the induction of anticancer effects mediated by suppression of the Notch signaling pathway [57], and the induction of a phenotype transformation towards non-tumorigenic cells, along with a decrease in proliferation and increase in differentiation, as well as apoptosis [58].

MRK-003, alone or combined with GSNO or chloroquine, reduced the baseline side population in primary glioma cultures and suppressed the increase of the side population induced by GSNO [59]; prevented neurosphere formation in HCMV-infected GBM cells and reduced the functionality or number of CSCs [60]; decreased the viability and sphere-formation capacity and increased apoptosis through suppression of the Akt pathway [61]; and induced autophagy in glioma neurosphere lines and reduced cell proliferation, cell growth, and the colony formation ability [62].

GSI-I treatment sensitized U251 and U87 cell lines to radiation through the reduction of radio-resistant CD133^+^ cells, enhanced the radio-sensitivity in cancer cells, and suppressed the tumor growth [63]. GSI-I also enhanced the therapeutic effect of temozolomide and led to an increase in CD133^+^ glioma cytotoxicity [64].

In a study by Pietras et al. [65], MK-003 (10 μM), alone or in combination with tetradecanoyl phorbol acetate, suppressed the glioma primary cells induced by PDGF and eliminated the cancer cells expressing stem cell markers.

In GSCs, RO4929097, either alone or in combination with farnesyltransferase inhibitors, blocked the Akt pathway and inhibited the cell-cycle progression, thus enhancing the radio-sensitivity and reducing the tumor growth. This combination, in addition to radiation, led to a durable response in orthotopic tumor models [66].

Treatment with MK0752 (25 μM) decreased the proliferation and self-renewal ability of GSCs and reduced the number of secondary neurospheres by differentiating GSCs into less proliferative glioma progenitor cells [67].

Another study by Dai et al. [68] investigated the change of glycosylation patterns upon treatment with GSI in GBM CSCs. For this purpose, compound E was cultured with these cancer cells, resulting in a phenotype transformation of CSCs toward a less tumorigenic form upon compound E treatment. Moreover, GSI-II treatment (0.2 µg) for 20 days effectively suppressed the CSC generation in U87 cells and significantly abrogated the proliferation and differentiation of U87 tumor-initiating cells [69].

### 5.4. Breast Cancer

The outcomes of seven in vitro and in vivo studies on breast cancer cells indicated that suppression of the Notch pathway with DAPT suppressed the activation of Notch by integrin-linked kinase (ILK). ILK knockdown blocked breast CSCs in vitro [70]; reduced the expression of Notch signal effectors NICD, Jagged 1, HES1, and signal peptide CUB EGF-like domain-containing protein 2 (SCUBE2); and decreased the self-renewal ability of breast CSCs [71]. It also blocked the cleavage of the CD44 intracytoplasmic domain, reduced the mammosphere-forming ability, decreased cell invasion and proliferation of triple-negative breast cancer (TNBC) cells, and suppressed tumor formation in mouse xenograft models [72,73]. These studies also showed that treatment with DAPT decreased the number of mammary progenitor cells and stem cells in p53-deficient mammary epithelium [74], reduced the percentage of CD44^hi^/CD24^lo^ cells, lessened micro- and macro-metastases in mice, and inhibited the colony formation ability of brain metastatic MDA-MB-231 cell lines [75]. Moreover, DAPT treatment led to a significant inhibition of Notch-mediated cell survival and invasion under hypoxia through increasing E-cadherin expression and suppressing the phosphorylation of Akt in breast cancer cells [76].

Five other studies reported that treatment with various concentrations of DAPT, alone or combined with different factors (e.g., radiation, lapatinib, gefitinib, tamoxifen, and 6-shogaol), led to inhibition of the mammosphere-forming ability and an increased TIC gene expression signature, and suppressed the expansion of CD44^+^CD24^low+^ TRCs after radiation [77], decreased mammosphere formation and the acini size in DCIS cell lines by inhibiting Notch and ErbB1/2 [78], blocked CSC activity induced by estrogen both in vitro and in vivo, and helped gefitinib to entirely suppress the estrogen effect [79]; inhibited the estrogen receptor-α promoter activity, reduced the tamoxifen sensitivity, and enhanced the expression of markers associated with basal-like breast cancers [80]; and blocked the spheroids and breast cancer cell proliferation and suppressed the colony formation capacity and number of spheroids through inhibiting the Notch pathway [81].

In a study by Mamaeva et al. [82], the anticancer effect of glucose-functionalized nanoparticles carrying DAPT on breast CSCs was assessed. To induce Notch suppression, the mesoporous silica nanoparticles were loaded with the compounds. In vitro, breast CSCs were exposed to variable concentrations of DAPT nanoparticles (5–50 μg/mL) for 24 h, indicating that DAPT treatment reduced the CSC population. In vivo, treatment with DAPT-loaded particles or free DAPT led to a significant decrease in the size of the cancer cell population. As a result, glucose-functionalized mesoporous silica nanoparticles carrying DAPT significantly eliminated the number of CSCs.

GSI-XVII therapy (5 μM, with or without radiation) decreased the self-renewing capacity and prevented the recombinant human erythropoietin-induced enhancement in primary sphere formation [83], while preventing the radiation-induced DLL3, Notch2, Jagged1, and DLL1 gene expression and significantly decreased the number of breast CSCs [84]. It was reported that various concentrations of GSI-I, ranging from 1–10 μM for 24, significantly inhibited breast CSCs [85].

The in vitro and in vivo treatment of different concentrations of MRK-003, either alone or in combination with lapatinib, trastuzumab, and docetaxel, resulted in Notch-1 inhibition, prevented mammosphere formation, inhibited the proliferation of bulk HER2^+^ HCC1954 cells, and prevented tumor relapse in xenograft models [86], and suppressed Notch signaling activation and decreased the number of CSCs, while a combination of MRK-003 and docetaxel enhanced this activity [87]. MRK-003 was also shown to decrease the viability of cells derived from tumorspheres in vitro, reduce tumor-initiating cells, and block the proliferation and self-renewal ability of mammosphere-resident cells, and induced their apoptosis and differentiation [88].

Two studies using different concentrations of DAPT and compound E with or without AD-01 reported that the treatment inhibited the growth and metastasis of the cancer stem-like cells of 231BrM in the brain through suppressing the expression of HES5 in vitro and in vivo [89]. In addition, the combined treatment of DAPT and compound E with AD-01 led to a reduction in the mammosphere-forming efficiency (MSFE) and enhanced the anti-CSC effects [90].

In another study, mammosphere or monolayer breast cancer cell cultures were treated with DAPT (10 μM), and MCF-7 cells were treated with dibenzazepine (DBZ) (10 μM) for 3 days. DAPT treatment decreased the proportion of ESA^+^/CD44^+^/CD24^low^ cells. Both DAPT and DBZ significantly reduced the N1-ICD and decreased the HEY2 and HES1 expression. In an in vivo study, mice were treated with 1 mg/mL DBZ for 3 days. DBZ treatment completely ablated tumor initiation and significantly decreased the size and volume of MCF-7 tumors [91].

The administration of LY-411 and LY-575 (25–50 μM), MRK003 (10–20 μM), and LLNle (0.5 μM) for 1–28 days significantly decreased the mammosphere-forming ability. MRK003 treatment irreversibly inhibited the mammosphere-forming ability, but treatment with either Ly-411,575 or LLNle had a transient effect on mammosphere formation [92].

MK-0752 (25 μM) and RO4929097(10 μM), alone or in combination with tocilizumab, suppressed tumor growth, but enhanced the CSCs in breast cancer cells expressing Notch3, while inducing IL-6. Treatment with MK-0752 led to the induction of IL-6 through Hey2-Notch3 signaling inhibition. Furthermore, hypoxia-inducible factor 1α (HIF1α) downregulated breast CSCs by reducing the IL-6 levels in breast cancer cells expressing Notch3. Using in vivo xenograft models, the concurrent use of tocilizumab and MK-0752 caused a significant reduction in breast CSCs and suppressed tumor growth [93]. Another study also indicated that MK-0752 treatment with or without docetaxel in mice bearing human tumorgrafts reduced the primary and secondary mammosphere-forming efficiency (MSFE); decreased the ALDH^+^ and CD44^+^/CD24^−^ subpopulations; downregulated NICD, Hes1, Hey1, Hes5, and Myc; and reduced tumor growth. Taken together, GSI treatment decreased breast CSCs and increased the efficacy of docetaxel [31].

The co-culture of CD44^+^CD24^low+^ and CD44^+^CD24^neg^ cells with different GSIs (RO4920927, 10 µM and DAPT, 5–10 µM) for 1–12 days led to a significant reduction in N1-ICD and decreased the expression of Sox2 and the sphere formation ability. Taken together, the blockade of Notch decreased the Sox2 expression and colony- and sphere-forming capacity. In in vivo nude mice, RO4920927 inhibited tumor growth in CD44^+^CD24^low+^ cells [94].

In another investigation, RO4929097 (0.1 nM–10 μM) treatment decreased the expression of Hey1, HeyL, and Hes1 and significantly reduced the colony-forming capacity of SUM149 and SUM190 cells irritated with radiation. In 2D and 3D clonogenic assays, RO4929097 treatment sensitized cell lines to ionizing radiation. The results also demonstrated that treatment with RO4929097 suppressed inflammatory cytokine synthesis, including interleukin (IL-8), IL-6, and tumor necrosis factor-α (TNF-α) [95].

### 5.5. Colorectal Carcinoma

DAPT treatment (10–20 μmol/L) led to the blockage of Notch signaling and partially suppressed the effect of KRAS on Hes1 in colorectal carcinoma cells. The DAPT treatment also reduced the number of CSCs [96].

In another study, the inhibition of Notch by DAPT significantly reduced the Lgr5-GFP cell population in Lgr5-EGFP-CreER^T2^ organoids. The suppression of Notch by GSI led to a reduction in Ascl2 levels and also enhanced apoptotic cells shed into the lumen [97]. Furthermore, the suppression of Notch by DAPT decreased the colon cancer stem cell (CCSC) population and enhanced the non-CCSC population. DAPT treatment also inhibited asymmetric division and decreased symmetric CCSC-CCSC division [98].

When utilizing soluble Jagged-1-Fc protein and DAPT in colorectal cancer cells, it was observed that the Notch-1 signaling pathway activates epithelial–mesenchymal transition (EMT)/stemness-associated proteins Slug, Smad-3, and CD44 by inducing the expression of Jagged-1. Treating the parental cells with DAPT decreased the proteins, such as Jagged-1, Smad-3, and CD44, and induced a significant reduction of Slug in the ICN1 cells [99].

The administration of JLK6 and DAPT led to a significant decrease in the number of colonspheres in SW620 cell lines [100]. Moreover, another study showed that GSI-X treatment suppressed endothelial cell conditioned medium—induced Notch signaling activation and CSC enrichment in HCT116 cells [101].

Treating mice bearing CRC cells with PF-03084014 (125 mg/kg), alone or combined with irinotecan, for 28 days resulted in a significant reduction in tumor recurrence and tumor growth and also decreased the ALDH^+^ population. This combination therapy had a greater antitumor effect when compared to PF-03084014 or irinotecan alone [102].

### 5.6. Gastric Cancer

Hayakawa et al. [103] presented a study examining the effect of Notch signaling inhibition on gastric CSC elimination. For this purpose, organoids were co-cultured with 25 μM DAPT for 10 days. The addition of a GSI suppressed the growth of the corpus organoid. For in vivo experiments, the Notch inhibitor dibenzazepine was injected intraperitoneally (30 μmol/kg) into mice for 2 weeks. Dibenzazepine treatment decreased the Mist1-lineage tracing expansion and proliferation in the isthmus.

In another study, the CD44^+^ population of gastric CSCs was targeted by DAPT, both in vitro and in vivo. In gastric CSCs, DAPT (2.5–15 µM) treatment for 24–96 h decreased the size of the CD44^+^ population in a time- and concentration-dependent way. This study claimed that DAPT treatment led to the inhibition of the invasion, migration, and proliferation of CD44^+^ CSC [104].

Epithelial–mesenchymal transition (EMT) is a process associated with tumor initiation, invasion, metastasis, and resistance to therapy. The exposure of CD44^−^ and CD44^+^ cell lines with DAPT (10 μM) for 72 h suppressed the EMT markers and Hes1 expression, inhibited the CSC properties, and blocked the CD44^+^ cell proliferation and invasion. In addition, in vivo GSI treatment significantly suppressed the growth of CD44^+^ cell xenograft tumors [105].

The co-culture of CS12 and MKN45-133^+^ cells with DAPT (5μM) for a day resulted in the suppression of Notch1 activation and enhancement of CD133 and stemness genes. DAPT also inhibited the sphere-forming ability and decreased the size and number of spheres [106].

DAPT (25–50 µM) alone or in combination with cisplatin reduced the gastric cancer cell viability and decreased the number of CD44^high^Lgr-5^high^ cancer cells, representative of CSC properties. Treatment with GSI alone did not affect cell proliferation [107].

### 5.7. Head and Neck Cancer

In a study by Chen et al. [108], the administration of DAPT (100 μM), alone or in combination with cetuximab and cisplatin, suppressed the viability of OECM1 cells. DAPT treatment or Kruppel-like factor 4 (KLF4) knockdown led to a significant reduction in the number of KLF4^+^/CD44^+^ cells in overexpressing Twist1 OECM1 cells. The combination of DAPT and cetuximab exerted an antitumor effect on xenograft models of head and neck cancers.

The addition of DAPT (1–10 μM), alone or in combination with 5-fluorouracil (5-FU), to esophageal adenocarcinoma OE33 cells inhibited their growth. For in vivo studies, animals were treated with 20 mg/kg DAPT for up to 10 weeks. DAPT treatment suppressed tumor growth, decreased HES1 expression and the level of NICD, and also led to the enhancement of apoptosis and reduction in cell proliferation in vivo. Taken together, Notch signaling inhibition sensitized cancer cells to chemotherapeutic agents and eliminated CSCs [109].

In 2011, Mendelson et al. [110] examined the role of Compound E in Barrett’s esophageal adenocarcinoma. For in vitro studies, cell lines were treated with TGF-β and Compound E (500 nM–5 μM) for 72 h. The results showed that the treatment only suppressed cell proliferation in BE3 cell lines with high Notch signaling and TGF-β dysfunction.

Another study demonstrated that the combination of DAPT (5–10 μM and 10–20 mg/kg) with chemotherapeutic agents (docetaxel, cisplatin, and 5-FU) led to a synergistic increase in chemotherapy-enriched head-neck CSCs, both in vitro and in vivo. Taken together, the inhibition of NOTCH1 signaling reduced the tumor self-renewal capacity and number of CSCs and also decreased transcription factors of self-renewal and markers related to CSCs [111].

The results of another study indicated that NOTCH inhibition (GSI XXI, 5–10 μM) resulted in inhibition of the spheroid-forming ability and also inhibited the survival, migration, and transformation of head and neck squamous cell carcinoma cells [112].

### 5.8. Liver Cancer and Cholangiocarcinoma

It was demonstrated that treatment with 10 μmol/L of different GSIs (L-685,485 and DAPT) inhibited the growth of the epithelial cell adhesion molecule (EpCAM)-positive fraction in hepatocellular carcinoma cells, and in vivo Notch suppression caused significant antitumor effects in hepatomas, showing that Notch inhibition could block the stem cell properties of hepatic cancer cells [113].

In another study, PF-03084014 suppressed the self-renewal ability and proliferation of CSCs. It also decreased the tumor growth in vivo and inhibited the metastasis of hepatocellular carcinoma to the lung [114].

Cao et al. [115] showed that Notch signaling independent of CSL (CBF1, suppressor of hairless and LAG1) might have an important role in hepatic CSCs, and MRK003 treatment significantly suppressed the sphere-forming capacity and reduced the size of the human stem-like hepatocarcinoma cell population. Similarly, sorafenib and PF-03084014 suppressed the self-renewal ability and proliferation of hepatocellular carcinoma spheroids [116].

One study showed that Notch pathway inhibition by either miR-34a overexpression or treatment with DAPT suppressed the growth and colony-forming ability of human cholangiocarcinoma cells. The results also indicated that targeting miR-34a combined with DAPT is an effective treatment for cholangiocarcinoma [117]. Additionally, the proportion of CD24^+^CD44^+^ cells, colony-forming capacity, and mice tumorigenicity were suppressed. A combination of GEM and GSI significantly decreased viable TFK-1 and RBE cells compared with GEM alone [118].

### 5.9. Lung Cancer

Five studies revealed that DAPT, alone or in combination with cisplatin, suppressed the Notch pathway, reduced the number of primary pulmospheres, decreased the expression of the Notch pathway target genes (Hey1 and Hes1), and reduced the self-renewal capacity of primary spheres [119]; decreased the number of CD133^+^ cells induced by cisplatin and enhanced the sensitivity to doxorubicin and paclitaxel [120]; decreased the number of CD44^+^/CD24^−^ cells and inhibited the growth capacity of lung CSCs [121]; significantly decreased the ALDH^+^ lung cancer cells, and led to cell-cycle arrest, depletion in the number of ALDH^+^ cancer cells in a concentration-dependent manner, and a reduction in tumor cell proliferation and clonogenicity [122]; and suppressed the proliferation of CD133^−^ and CD133^+^ cells and had a small effect on the cell cycle [123].

It was demonstrated that RO4929097 (1–10 μmol/L) with or without cisplatin modulated the self-renewing activity of LCSCs via p-STAT3 and HES1 in human non-small cell lung cancer (NSCLC) cells, whereas RO4929097 treatment increased their platinum resistance. Moreover, treatment with RO4929097 inhibited the self-renewal of LCSCs and increased the platinum sensitivity, both in vitro and in vivo [124].

In another study, the exposure of MRK-003 (5–20 μM), with or without docetaxel, inhibited the sphere formation and self-renewal ability and decreased the NICD2 expression. In a mouse tumor xenograft model, MRK-003 reduced the expression of downstream effectors of Notch signaling [125].

PF-03084014 (1 μM) treatment of lung cancer cells, either alone or in combination with erlotinib, resulted in the elimination of the erlotinib-induced stem-like cell population by reducing Notch signaling activity. The inhibition of Notch3 and EGFR receptors decreased the stem-like cell population expansion [126].

Ali et al. [127] reported that in vitro and in vivo treatment of lung adenocarcinoma cells with DBZ, with or without auranofin, led to the inhibition of oncosphere growth, cell viability, soft agar growth, and significantly inhibited tumor growth. This combination therapy resulted in a significantly higher level of inhibition compared with either compound alone.

In vitro treatment with DFPAA decreased the differentiation of NG2+ cells (Sca1hi/CD146-/CD45-/CD31-) into pericytes and reduced the number of neoplastic cells [128].

### 5.10. Melanoma

In 2016, a study by Kumar et al. [129] investigated the effects of targeting Notch1 on CSC-mediated melanoma progression. Treatment with DAPT and L-685,458 suppressed the expression of CD133 in CD133^+^ cells and enhanced the number of CD133^−^ cells. The results of this study demonstrated that inhibition of the Notch signaling pathway led to CD133-dependent mitogen-activated protein kinase (MAPK) signaling inhibition and eventually increased the interaction of tumor-associated endothelial cells and the migration of CD133^+^ cells, and blocked metastasis, melanoma growth, and angiogenesis.

It has also been shown that the Notch pathway has an important role in the differentiation of tumor pericyte precursors. DFPAA treatment led to a reduction in the number of NG2^+^ cells (oligodendrocyte precursor cells or polydendrocytes) [128].

The effect of the combination of GSIs with a Bcl-2 inhibitor on killing melanoma-initiating cells was investigated in a study. For both in vitro and in vivo studies, GSI-I (0.83 μM) was used, either alone or in combination with ABT-737, for 24–21 days. Combination therapy decreased the cell viability and promoted the non-melanoma-initiating cell apoptosis, inhibited primary sphere formation, decreased the number of ALDH^+^ cells, and suppressed the melanoma-initiating cells’ self-renewability ability in vitro. In a mouse xenograft model, combination therapy caused a significant decline in the tumor-initiating capacity [130].

In melanoma cancer stem-like cells (MCSLCs), DAPT (10 μM) treatment increased the expression of E-cadherin and inhibited the expression of VE-cadherin and Twist1, demonstrating that DAPT plays an important role in inhibiting melanoma metastasis [131].

### 5.11. Osteosarcoma

Yu et al. [132] investigated the role of Notch inhibition in the elimination of osteosarcoma stem cells (OSCs). OSCs were treated with 20 μM GSI (DAPT and RO4929097) for 24 h. GSI pretreatment suppressed the spheroid-forming ability and blocked the activity of cisplatin-enriched OSCs. The administration of GSI (DAPT, 10 mg/kg/d) for 14 days in mice bearing chemoresistant xenograft tumors reduced the sarcosphere-forming ability and suppressed the recurrence of the tumor. GSI treatment also downregulated the expression of stem-like cell markers.

In another study, different osteosarcoma cell lines were exposed to GSI (DAPT, 0–50 μM), alone or in combination with cisplatin, for 8–72 h. Moreover, nude mice were treated with 8 and 10 mg/kg DAPT, intraperitoneally (i.p.), alone or combined with cisplatin (CDDP), for 5 weeks. The study suggested that GSI treatment increased the anticancer effect of CDDP in resistant OSCs through the inhibition of proliferation, reduction in motility, induction of apoptosis, and cell-cycle arrest. In addition, GSI treatment reduced the number of OSCs and enhanced the platinum sensitivity of the tumor. It was also shown that the cotreatment of GSI and CDDP suppressed phosphorylated extracellular-regulated kinase (ERK) and Akt, and thus enhanced the antitumor effects. In mice, combination therapy exhibited a greater effect on suppressing the CDDP-resistant tumor xenograft metastasis and growth, compared with the compound alone. Taken together, the GSI and CDDP combination sensitized CDDP-resistant human osteosarcoma cell lines to CDDP through Notch signaling downregulation [133].

### 5.12. Ovarian Cancer

Vathipadiekal et al. [134] investigated the effect of DAPT on the elimination of ovarian cancer side population cells. The SKOV3 SP and MP cells were cultured with 10 and 20 µg DAPT, for 8 days. DAPT treatment repressed the colony-formation and survival of ovarian cancer side population cells. This side population had a concentration-dependent sensitivity to DAPT treatment.

In another study, primary ovarian tumor cells were exposed to GSI-I (1–10 μM), alone or in combination with cisplatin, for 1–2 days. GSI-I decreased the CSC population and enhanced the tumor platinum-sensitivity. The knockdown of Notch3 using small interfering RNAincreased the platinum therapy response, demonstrating that tumor chemo-sensitivity modulation by GSI-I is Notch signaling specific. In addition, the study demonstrated that the combined treatment of DAPT and cisplatin had a synergistic antitumor effect in Notch-dependent cancer cells through increasing the G(2)/M cell cycle arrest, apoptosis, and DNA-damage response [135].

The treatment of ovarian cancer stem-like cells with DAPT (1–20 μM, 1–3 days) showed that the Notch blockade with DAPT significantly hampered ovarian CSC proliferation and the self-renewal ability, reduced the ovarian cancer stem cell (OCSC)-specific surface marker expression, and decreased the mRNA and protein expression of Sox2 and Oct4 in OCSCs [136].

Recently, the effect of Notch3 signaling pathway suppression was investigated in NR2F6-overexpressing epithelial ovarian cancer (EOC) in vitro and in vivo. RO4929097 (10 μM treatment) enhanced the stem cell phenotype and increased the CDDP resistance in EOC cells by inducing Notch3 activation. RO4929097 treatment also inhibited ovarian cancer cell proliferation and increased apoptosis [137].

### 5.13. Pancreatic Carcinoma

In pancreatic cancer cells, DAPT-induced Notch downregulation led to a reduction in the number of CD133^+^, the inhibition of cell proliferation, and abrogation of the DLL4/Notch-induced chemo-resistance [138], and also resulted in apoptosis, the inhibition of EMT, and the suppression of tumorigenesis by eliminating pancreatic cancer-initiating cells (CD44^+^/EpCAM+), both in vitro and vivo [139].

In four different studies, the administration of 1–80 μM DAPT, alone or in combination with leptin, gemcitabine (GEM), and 5-FU, inhibited the Notch pathway, reduced the proliferation of MiaPaCa and BxPC-3 cells, decreased the number of leptin-induced CD133^+^ and ALDH^+^ cells, and inhibited tumorsphere formation [140]. In addition, the number of CM and insulinomas (INS) CSC-enriched spheres decreased, whereas the INS CSC-like cells’ sensitivity to 5-FU improved and the clonogenicity and tumorigenicity of INS CSC-like cells were decreased [141]. The proportion of CD24^+^CD44^+^ cells and pAkt, Hes1, and β-catenin expression in cell lines treated with gemcitabine declined, and the invasion and migration abilities were reduced [142].

DAPT (10–100 nM) and MRK-003 (0.72–5 μM) treatment decreased the number of AcTub^HI^ cells, and in vivo, significantly decreased the abundance of mPanIN epithelial cells expressing Dclk1, which was correlated with blockade of PanIN progression [143].

It was reported that the in vitro and in vivo treatment of cells with different GSIs (MK-0752 and RO4929097), with or without gemcitabine, led to a decrease in the number of CSCs and the inhibition of tumorsphere formation and blocked tumor growth in NOD/SCID mice. Treatment with MK-0752 or RO4929097 combined with gemcitabine displayed the highest percentage of apoptosis compared with either compound alone [144].

MRK-003 with or without gemcitabine downregulated the intracellular domain of the Notch protein (NICD), eliminated the CSCs, and inhibited the colony-forming capacity in pancreatic cancer cells. The combination of gemcitabine and MRK-003 enhanced the antitumor effects, reduced tumor cell proliferation, and led to the induction of intratumoral necrosis and apoptosis [145].

Treating mice bearing pancreatic cancer xenografts with PF-03084014 (150 mg/kg), alone or combined with gemcitabine, downregulated NICD Hey-1 and Hes-1, resulted in tumor regression, and reduced cancer stem cells. The results also showed that combination therapy with gemcitabine and PF-03084014 was effective in apoptosis induction, the suppression of angiogenesis, and tumor cell proliferation, leading to a reduction of primary tumor growth, as well as controlling metastatic dissemination, compared to treatment with gemcitabine alone [146].

### 5.14. Prostate Cancer

In a recently published study, Wang et al. [147] investigated the antitumor effect of PF-03084014 on prostate cancer. Prostatic cancer stem cells were exposed to 1–100 μM PF-03084014 in combination with oncolytic herpes simplex virus for 6 days. The results of this study showed a reduction in the cancer cell population. 

It was shown that DAPT (1 nM–400 μM) treatment reduced the protein levels of NICD1 and blocked γ-secretase activity in prostate cancer cells [148].

In castration-resistant prostate cancer (CRPC), PF-03084014 (0.1–10 μM) treatment, with or without docetaxel, resulted in a greater decrease in the tumor growth of both docetaxel-resistant and docetaxel-sensitive CRPC compared with either compound alone. The results also showed that this treatment increased the anticancer activity and apoptosis, decreased the epithelial to mesenchymal transition, reduced survival signals (EGFR, cyclin E, NF-κB, and PI3K/Akt pathway; Bcl-xL; and Bcl-2), decreased the density of microvessels, and eliminated CSCs in the tumor [149].

Various studies, as presented in this review, have demonstrated the anti-CSC effect of GSIs. GSIs have different chemical structures, but despite these differences, all of them can inhibit Notch signaling, making them excellent candidates for potential anti-CSC agents. The anti-CSC effect of GSIs has been tested in various cancer cell lines and animal models. Various GSIs, with different concentrations, doses, and treatment durations, have been used in the reviewed studies. The elimination of CSCs using GSIs depends on the GSI type, cell line used, GSI concentration, and other anticancer agents used for combination treatment. In vitro anti-CSC activity and in vivo tumor growth inhibition accompanied by longer survival provides the proof of the efficacy of GSIs. All reviewed studies indicate consistency of their results and confirm Notch inhibition by using GSIs.

Probable mechanisms involved in observed anti-CSC effects of GSIs include apoptosis, cell cycle arrest, reduced survival signals (EGFR, cyclin E, NF-κB, and PI3K/Akt pathway; Bcl-xL; and Bcl-2), decreased protein levels of NICD1 mRNA, and protein expressions of Sox2 and Oct4 [135,136,137,138,139,144,145,148,149]. Most studies compared results obtained with GSIs in the presence and absence of other anticancer drugs. The results of these studies showed that GSIs in combination with other chemotherapeutic drugs have a greater inhibitory effect on CSCs. Additionally, GSIs increase the sensitivity of CSCs to anticancer drugs and enhance their anti-CSC effects [48,49,50,51,52,144,145,146].

The reviewed studies showed that increased concentrations of GSIs reduce the CSC growth rate and their anti-CSC effects are time- and concentration-dependent [36,54,104,122,134]. Higher concentrations of GSIs have a higher risk of toxicity. Therefore, a balance between toxicity and enhancement of the concentration should be set. One study showed that the combination of GSI with glucose-functionalized nanoparticles increased the anti-CSC effect and, at the same time, required a lower concentration to effectively reduce the CSCs in breast cancer [82]. If GSI-loaded nanoparticles are selectively localized in tumor cells, cancerous tissue can receive a higher dose in comparison to normal tissue. Therefore, the toxicity and side effects of GSI could be reduced.

## 6. Conclusions

There is evidence that CSCs induce tumor perpetuation, even after effective therapies, and result in aggression of the tumor. Given that Notch signaling is implicated in regulation of the cell fate, the aberrant activation of this pathway can lead to tumorigenesis. Therefore, the suppression of this pathway could be a potential therapeutic target for cancer management. This is the first study that provides a systematic review of the literature on the mechanisms of action of GSIs against various CSCs, although there are many other studies on the effect of GSIs on cancer cells rather than CSCs. Various studies have revealed that GSIs have potent anti-CSC effects and can inhibit the neoplastic activities, angiogenesis, and tumor growth of cancer cells, especially when combined with chemotherapeutic or targeted therapy drugs. Since Notch signaling has a critical role in the self-renewal and maintenance of CSCs, GSIs have a greater effect on CSCs in comparison to cancer cells. Therefore, the eradication of CSCs by GSIs could lead to higher survival rates.

The potential ability of GSIs to target CSCs is mediated through the inhibition of various stemness-related signaling pathways and transcription factors. Of note, the importance of the other cellular events and signaling pathway interactions that contribute to tumor progression should be considered when it comes to designing a therapeutic plan that involves GSIs. Apart from their advantages, GSIs cause adverse severe side effects. For some GSIs, a high potency and optimal physical properties have been achieved, but the biological mechanism imposes inbuilt Notch-related side effects. Therefore, they are dose-limiting and need moderate, intermittent administration. Consequently, the effective inhibition of Notch pathway activity requires more targeted delivery approaches and the effective delivery of GSIs to CSCs to target Notch signaling in this population. One of the strategies for overcoming these challenges is using the specific properties of CSCs in the design of nanoparticles containing GSIs to improve the outcome of targeted treatment. Another strategy could be the use of small molecules, such as CB-103, which might target the Notch transcription complex to downregulate Notch signaling. In this systematic review, we conclude that inhibiting Notch signaling by GSIs is a promising strategy for achieving efficient cancer therapy.

## Figures and Tables

**Figure 1 molecules-26-00972-f001:**
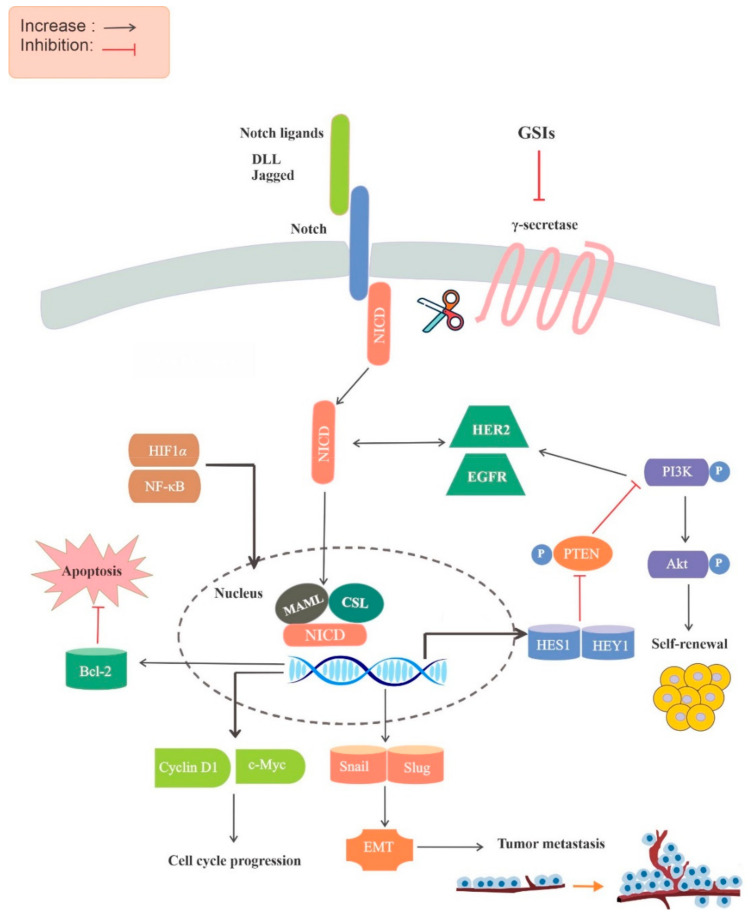
Molecular mechanisms underlying the anticancer properties of GSIs. First, the Notch receptor interacts with the Notch ligand, such as DLL, and initiates proteolytic cleavage at the extracellular site, followed by cleavage at the intracellular site by γ-secretase, leading to the release of NICD. Then, NICD is translocated into the nucleus, where it interacts with CSL and MAML to form a transcription-activating complex. GSIs can inhibit these steps, including receptor/ligand binding, the release of NICD, and the interaction of NICD and downstream targets, as well as NICD protein stability, and can thus have anticancer effects. Abbreviations: CSL, CBF1, suppressor of hairless and LAG1; DLL, Delta-like ligand; EGFR, epidermal growth factor receptor; EMT, epithelial-mesenchymal transition; GSIs, γ-secretase inhibitors; HER2, human epidermal growth factor receptor 2; HIF1α, hypoxia-inducible factor 1α; MAML, mastermind-like; NF-κB, nuclear factor-kB; NICD, Notch intracellular domain; PI3K, phosphatidylinositol-3-kinase; PTEN, phosphatase and tensin homolog.

**Figure 2 molecules-26-00972-f002:**
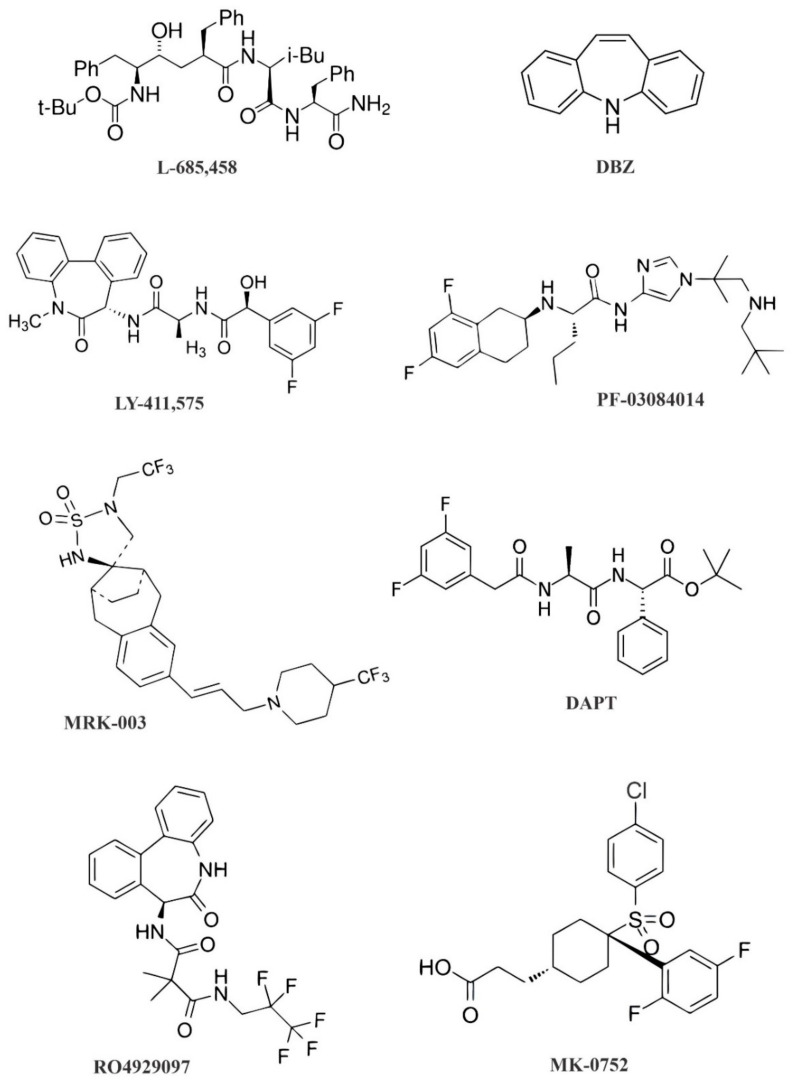
Chemical structure of selected γ-secretase inhibitors. Abbreviations: DAPT, (*N*-[*N*-(3,5-difluorophenacetyl)-l-alanyl]-S-phenylglycine t-butyl ester) and DBZ, dibenzazepine.

**Figure 3 molecules-26-00972-f003:**
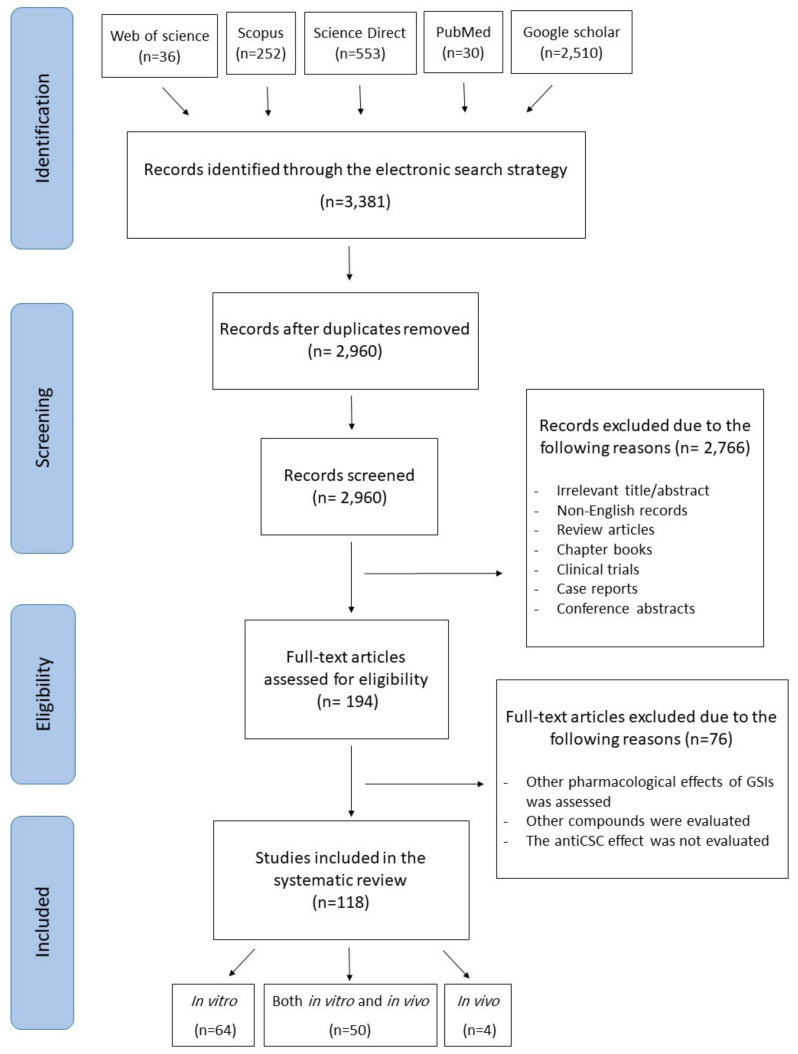
Chemical structure of selected γ-secretase inhibitors. Abbreviations: DAPT, (*N*-[*N*-(3,5-difluorophenacetyl)-l-alanyl]-S-phenylglycine t-butyl ester) and DBZ, dibenzazepine.

**Table 1 molecules-26-00972-t001:** Potential anti-cancer stem cell (CSC) effects and related mechanisms of action of γ-secretase inhibitors (GSIs) based on in vitro studies.

Cancer Type	Cell Type	Type and Properties of GSIs	Conc.	Exposure Time	Major Outcome	References
Adenoid cystic carcinoma	Accx11 cells	DAPT DAPT + radiation	1–10 μM	24–96 h	↓SKP2 and N1ICD, ↓growth of ACC, ↓CD133^+^ cells, ↑p27Kip, ↑radio-sensitivity	Panaccione et al. [33]
Blood (Lymphoma)	Lymphoma cells	L-685,458	0.1–100 μM	24 h	↓CSCs, ↓colony formation	Wang et al. [34]
Blood (Lymphoma)	EL4 and A20 cells 3D cell culture	DAPT DAPT + NSC23766	5 μM	24 h	↓CSCs, ↑sensitivity to doxorubicin	Ikram et al. [35]
Blood (Lymphoma)	Mono-nuclear cells (MNC)	DAPT	8–16 μM	14 days	↓CSCs, ↓colony formation ability, ↓proliferation	Gal et al. [36]
Blood (leukemia)	Tal1/Lmo2 leukemic cells	MRK-003		48 h	↓leukemia-initiating cells	Tatarek et al. [37]
Blood (leukemia)	DND-41, KOPT-K1, Jurkat, NB4, HL60, and OCI/AML-3 cells	Compound E Cy + Qu + Compound E	10 μM	1–7 days	↓growth, ↓clonogenicity, ↓NOTCH1, ↓CSCs	Okuhashi et al. [38]
Blood (leukemia)	AA and HEL cell lines	DAPT GSI-XII Compound E	20 μM (DAPT) 5 μM (GSI-XII) 10 μM (Compound E)	1–7 days	↓cell growth, ↓colony formation ability, ↑differentiation, ↓CSCs, ↑hemoglobin	Okuhashi et al. [39]
Blood (Multiple myeloma)	Human MM-CSCs	RO4929097 RO4929097 + Bruceantin	10 µM	24 h	↓CSCs, ↓viability, ↓migration, ↓proliferation, ↓angiogenesis	Issa et al. [40]
Brain (Glioblastoma)	GSCs	DAPT	25 μM	7 days	↓proliferation, ↓tumor spheres, ↓CD133 and GLAST, ↓tumor propagation	Hu et al. [41]
Brain (Glioma)	U87-MG and LN-22 cell lines	DAPT	1 μM	24 h	↓sphere-forming capacity, ↓N1ICD and SOX2 expression	Li et al. [42]
Brain (Glioma)	SHG-44 cell line	DAPT	0.5–10 μmol/L	1–5 days	↓proliferation, ↓CSCs, ↓CSC of CD133^+^	Liu et al. [43]
Brain (Glioblastoma)	GBM-derived CD105+ cells	DAPT	5 μM	48 h	↓transition between CD133^+^/CD144^−^ and double-positive, ↓CSCs	Wang et al. [44]
Brain (Glioblastoma)	Human-derived GBM xenograft cells	DAPT	1–10 µM	3–14 days	↓cell growth, ↓stem cell-like features, ↓CSCs	Kristoffersen et al. [45]
Brain (medulloblastoma)	Medulloblastoma-derived cells	DAPT	10 μM	8–72 h	↓HIF-1α and hes1, ↓nestin^+^ cells, ↓CSCs, ↑β-III-tubulin^+^ cells, ↑neuronal differentiation	Pistollato et al. [46]
Brain (Glioma)	029 and 036 neurosphere cultures Human-derived GBM xenograft cells	DAPT	10 µM	24 h	↓brain CSCs	Kristoffersen et al. [47]
Brain (Glioma)	ihBTC2, SU3, SU3-5R, and C6 cells	DAPT DAPT + radiation	2 μm	24 h–8 days	↑radio-sensitivity, ↑apoptosis, ↓CSCs	Yuntian et al. [48]
Brain (Glioma)	A172 cell line	DAPT DAPT + Gleevec + amph1D peptide	2/5–25 μm	24–48 h	↓CSCs, ↑cell death	Gal et al. [49]
Brain (medulloblastoma)	D341 cells	DAPT DAPT + HBMEC/ D341Med	2 μM	48 h	↓Bmi-1, CDK6, c-MYC, and CCND1 expression, ↓CD133^+^ cells, ↓CSCs	Wang et al. [50]
Brain (Glioma)	LN229 and U251 cell lines and primary cells isolated from GBM10 xenograft PDX models	DAPT DAPT + imatinib	20 μM	1–2 weeks	↓growth inhibition, ↓GSCs	Kanabur et al. [51]
Brain (Glioma)	U251MG, U87MG, T98G, A172, and U373MG Adherent GBM cell lines 0308 and 0822 GBM stem cell lines	DAPT GSI-loaded MLs DAPT + INCB3619	12/5–100 μM	1–6 days	↓HES1 and HEY1, ↓YKL-40/CHI3L1, ↓CSCs	Floyd et al. [52]
Brain (Glioblastoma)	GSCs	L-685,458 or DAPT	5 μM (L-685,458) 10 μM (DAPT)	3, 7, 14 days	↓GSCs, ↓sphere-forming cells	Park et al. [53]
Brain (Glioma)	U251, U87, A172, and LN18 cells	DAPT BMS-708163 RO4929097 RO4929097 + BMS-708163	0.5–18 μM	72 h	↓CSCs	Saito et al. [54]
Brain (Glioma)	T3359, T3691, T4105, T4302, and T4597 cells	DAPT L685,458	2 μM (DAPT) 0.5 μM (L685,458)	4 h–3 weeks	↑apoptosis, ↓growth, ↓clonogenic survival of GSCs	Wang et al. [55]
Brain (Glioblastoma)	GBM neurospheres	GSI-18 MRK-003	2 μM (GSI-18) 2–10 μM (MRK-003)	2–5 days	↓CSCs, ↓proliferation, ↑cell death, ↓STAT3 and AKT phosphorylation	Fan et al. [56]
Brain (Medulloblastoma)	DAOY, PFSK, D283Med, and D425Med cell lines	GSI-18	2 μmol/L	48 h	↓Hes1, ↓mRNA levels, ↓clonogenicity, ↓CSCs	Fan et al. [57]
Brain (Glioblastoma)	GBM neurosphere cultures	GSI-18	2–50 μM	48 h	↓proliferation, ↑apoptosis, ↑differentiation, ↓CSCs	Dai et al. [58]
Brain (Glioma)	T98G cell lines, human tumor neurospheres, and PDGF-induced glioma primary cells	MRK-003 MRK-003 + GSNO	3 μM	2, 6 h	↓side population cells	Charles et al. [59]
Brain (Glioblastoma)	Primary GBM cell lines	MRK003	2 μM	7 days	↓neurosphere formation, ↓CSCs	Fornara et al. [60]
Brain (Glioblastoma)	MD13, 30R, Me83, 1123M, 528P, 157NS, and 146NS cells	MRK003	1–10 μM	2–7 days	↓viability, ↓sphere-formation capacity, ↑apoptosis, ↓Akt pathway, ↓CSCs	Tanaka et al. [61]
Brain (Glioma)	HSR-GBM1 and JHH520 neurosphere lines	MRK003 MRK003 + Chloroquine	0.5–5 μM	48–72 h	↓proliferation, ↓CSCs ↑autophagy, ↓cell growth, ↓colony formation ability	Natsumeda et al. [62]
Brain (Glioma)	U87 and U251 cell lines	GSI-I with radiaton	1–5 μmol/L	4–72 h	↓CSCs, ↓CD133^+^ cells	Lin et al. [63]
Brain (Glioma)	U87MG cell line	GSI-I GSI-I + TMZ/cyclopamine	5 µmol/L	5 days	↓CD133^+^ cells, ↓CSCs, ↑TMZ therapeutic effect	Ulasov et al. [64]
Brain (Glioma)	PIGPCs	MK-003 MK-003 + TPA	10 μM	24 h	↓stem cell markers, ↓glioma primary cultures, ↓CSCs	Pietras et al. [65]
Brain (Glioma)	UAB-1005, 1051, 1027A, and 1079 cell lines	RO4929097 RO4929097 + Farnesyltransferase	20–500 nM	24–72 h	↓AKT, ↓cell-cycle progression, ↑radio-sensitizing, ↓CSCs	Ma et al. [66]
Brain (Glioblastoma)	GSCs	MK0752	25 µM	1–7 days	↓self-renewal ability, ↓proliferation, ↓GSCs, ↓secondary neurosphere	Hu et al. [67]
Brain (Glioblastoma)	GBM neurosphere cultures	Compound E	-	-	↓CSCs	Dai et al. [68]
Brain (Glioma)	U87 cells	GSI-II	0.2 µg	20 days	↓CD133, ↓proliferation, ↓CSCs	Ding et al. [69]
Breast	MCF-7, MDA-MB-468, and MDA-MB-231 cells	DAPT	10 μM	12–and/or 24-h	↓CSCs, ↓ILK-induced Notch activation	Hsu et al. [70]
Breast	Human TNBC, Hs578T, and MDA-MB-231 cell lines	DAPT	1–5 μm	5–12 days	↓BCSCs, ↓sphere formation capacity, ↓self-renewal ability, ↓NICD, SCUBE2, HES1 and jagged 1	Chen et al. [71]
Breast	ZR-75–1, MCF-7, T-47D, JIMT, SK-BR-3, and MDA-MB-231 cell lines	DAPT	2 and 5 μM	12–24 h	↓mammosphere formation, ↓CD44ICD cleavage, ↓CSCs	Cho et al. [72]
Breast	HCC38 and HCC1806 stem cell lines	DAPT	10–40 µM	1–14 days	↓BCSCs, ↓mammosphere-forming ability, ↓cell invasion, ↓cell proliferation, ↑cell death	Li et al. [73]
Breast	Trp53−/−mammospheres and TM40A-let7^s^-p53KD mammospheres	DAPT	5 μM	7 days	↓mammary stem/progenitor cells	Tao et al. [74]
Breast	Athymic nude nu/nu mice bearing HCC1806 cells	DAPT	5 μmol/L	24–48 h	↓CSCs, ↓percentage of CD44^hi^/CD24^lo^ cells, ↓colony formation ability	McGowan et al. [75]
Breast	MCF7 and MDA231, MDA231LM, and MDA231BoM cell lines	DAPT	10–20 μM	24–72 h	↓Akt, ↓CSCs	Xing et al. [76]
Breast	CAF61a, HCC1937, THP-1, MDA-MB-231 (1833), MDA-MB-436, MDA-MB-231, Hs578T, MDA-MB-157, SKBR3, T47D, MCF7, HCC70, MDA-MB-468 cells	DAPT RT + DAPT	10 μM	48 h	↓mammosphere-forming ability, ↓CD44^+^CD24^low+^ TRCs, ↑TIC gene expression signature, ↓CSCs	Boelens et al. [77]
Breast	SUM225 and MCF10DCIS cell lines	DAPT DAPT + lapatinib/gefitinib	0.1, 5, 10, 20 µM	5, 7, 21 days	↓acini size, ↓mammosphere formation, ↓ErbB1/2	Farnie et al. [78]
Breast	T47D, MDA-MB-231, MCF7, BT474 cells	DAPT DAPT + Gefitinib	10 μM	24–48 h	↓CSCs, ↓oestrogen effect	Harrison et al. [79]
Breast	MCF-7 and T47D cell lines	DAPT DAPT + DSL/ Tamoxifen	0/5–1 μM	24 h	↓tumoursphere growth, ↓ER-α promoter activity, ↓CSCs, ↓tamoxifen sensitivity	Buckley et al. [80]
Breast	MCF-7 and MDA-MB-231 cells	DAPT DAPT + 6-shogaol	25–50 μM	1–7 days	↓CSCs, ↓proliferation, ↓colony formation capacity, ↓number of spheroids	Ray et al. [81]
Breast	MDA-MB-231 and MCF7 cells	DAPT DAPT loaded glucose-functionalized MSNs	1 μg	24 h	↓CSCs, ↓ALDH activity	Mamaeva et al. [82]
Breast	MCF-7, T47D, and MDA-MB-231 cell lines	GSI-XVII	5 μM	30 min	↓self-renewal, ↓CSCs, ↓primary sphere formation	Phillips et al. [83]
Breast	SUM159, MCF-7, and T47D cells	GSI-XVII GSI-XVII + radiation	5 μM	1–4 days	↓CSCs, ↓DLL3, Notch2, Jagged1, and DLL1 gene expression	Lagadec et al. [84]
Breast	HCC1937, HCC1806, MCF7, T47D, SUM149, and SUM159 cells	GSI-I	1–10 μM	24 h	↓BCSCs	Luo et al. [85]
Breast	MDA-MB-453, HCC1954 and MCF-7 cells	MRK-003 MRK-003 + lapatinib/trastuzumab	5 μM	7 days	↓CSCs, ↓mammosphere formation, ↓proliferation of bulk HER2^+^ HCC1954 cells	Shah et al. [86]
Breast	MCF7, MDA-MB 436, MDA-MB 231, ZR-75-1, ZR-75-30, and T47D cell lines	MRK-003 MRK-003 + docetaxel	10 mmol/L	1–3 weeks	↓breast CSCs, ↓self-renewal	D’Angelo et al. [87]
Breast	Primary tumor cells Mammospheres	MRK-003	0.01–10 μM	2–4 days	↓proliferation, ↓self-renewal ability, ↑apoptosis, ↑differentiation, ↓CSCs	Kondratyev et al. [88]
Breast	MDA-MB231, MDA-MB231BrM, CN34, and CN34BrM cells	DAPT	5–10 µM	48–72 h	↓CSCs, ↓HES5	Xing et al. [89]
Breast	ZR-75, MCF-7, and MDA-231 cell lines	compound E DAPT DAPT +AD-01	0.025–1.25 μmol/L (compound E) 10 μmol/L (DAPT)	72 h	↓BCSCs, ↓MSFE	McClements et al. [90]
Breast	MCF7, MDA-MB-231, and BT474 cells	DAPT DBZ	10 μmol/L	3–7 days	↓ESA^+^/CD44^+^/CD24^low^ cells, N1-ICD, ↓HEY2 and HES1, ↓CSCs	Harrison et al. [91]
Breast	MCF7, T47D-A18, T47D-C42, BT474, and SKBR3 cells	MRK003 GSI-I LY-411,575	GSI-I (0.5 μM), MRK003 (10–20 μM), LY-411, 575 (25–50 μM)	24–48 h 12–28 days	↓CSCs, ↓mammosphere formation	Grudzien et al. [92]
Breast	SUM149 and SUM159, MCF-7, MDA-MB-231, and HCC1954 cell lines	RO4929097	10 μM	7 days	↓BCSCs	Wang et al. [93]
Breast	MDA-MB-231-luc and MCF-10A cells	DAPT	5–10 µM	1–12 days	↓CSCs, ↓N1-ICD, ↓Sox2, ↓sphere formation ability	Azzam et al. [94]
Breast	SUM149 and SUM190 cells	RO4929097 RO4929097 + radiation	0.1 nM–10 μM	24–14 days	↓Hey1, HeyL, and Hes1, ↓colony-forming capacity, ↓CSCs, ↓TNF-α, IL-8, and IL-6, ↑sensitivity to ionizing radiation	G. Debeb et al. [87]
Cholangiocarcinoma	CCLP1, SG231, HUCCT1, and TFK1 cells	DAPT	10–40 μM	24–72 h 10–14 days	↓ growth, ↓colony formation, ↓CSCs	Kwon et al. [117]
Cholangiocarcinoma	HuCCT1, TFK-1, and RBE cell lines	DAPT DAPT + GEM	20–40 μM	1–4 days	↓CD24^+^CD44^+^ cells, ↓colony-forming capacity, ↓CSCs	Aoki et al. [118]
Colorectal	IEC-6/KRAS G12V cells	DAPT	10–20 μmol/L	24 h	↓CSCs, ↓Hes1	Feng et al. [96]
Colorectal	CCSC cells	DAPT			↓CCSCs, ↓symmetric CCSC-CCSC division, ↓asymmetric division, ↑non-CCSCs	Bu et al. [98]
Colorectal	HCT-116 cell line	DAPT	10 μM	1–10 days	↓CSCs, ↓Smad-3, Jagged-1, and CD44, ↓Slug	Fender et al. [99]
Colorectal	Caco-2 and SW620 cell lines	DAPT JLK6	2.5–10 µM (JLK6) 2.5–30 µM (DAPT)	1 h–14 days	↓colosphere formation, ↓CSCs	Moon et al. [100]
Colorectal	HCT116 cells	L-685,458			↓CSCs, ↓NICD upregulation, ↓Aldefluor-positive cell population	Lu et al. [101]
Gastric	Human gastric cancer cell lines, corpus gland cultures, and Mist1^+^ stem cells	DAPT	25 μM	10 days	↓Mist1^+^ stem cell proliferation	Hayakawa et al. [103]
Gastric	MKN45 cell line	DAPT	2.5–15 µM	24–96 h	↓CSCs, ↓proliferation, ↓CD44^+^ cells	Barat et al. [104]
Gastric	MKN-45 cells	DAPT	10 µM	72 h	↓CSCs, ↓EMT markers, ↓Hes1, ↓proliferation	Li et al. [105]
Gastric	GI2, CS12, MKN45 cells	DAPT	5 μM	24h	↓sphere-forming ability, ↓sphere size and number, ↓stemness genes and CD133	Hong et al. [106]
Gastric	SC-6-JCK, SH-10-TC, MKN74, and MKN45 cells	DAPT + Cisplatin	25–50 µM	24–72 h	↓CSCs, ↓cell viability, ↓CD44highLgr-5high population	Kato et al. [107]
Head and neck	SAS, OECM1, and FADU cells	DAPT DAPT + Cetuximab	100 μM	1–14 days	↓KLF4^+^/CD44^+^ cells	Chen et al. [108]
Head and neck	OE33, OE19, FLO1, JH-EsoAd1 cells	DAPT DAPT + 5-FU	1–10 μM	1–7 days	↓CSCs, ↓HES1 expression, ↓NICD, ↑apoptosis, ↓proliferation, ↑sensitivity to chemotherapeutic agents	Wang et al. [109]
Head and neck	FLO-1, SKGT-4, BE3, and OE33 cells	Compound E	500 nM–5 μM	72 h	↓CSCs, ↓proliferation	Mendelson et al. [110]
Head and neck	CAL27 and FaDu cell lines	DAPT DAPT + chemotherapeutic agents	5–10 μM	1–14 days	↓CSCs, ↓CSCs markers	Zhao et al. [111]
Head and neck	AW13516, NT8e, CAL27, DOK cells	Compound E	5–10 μM	20 h–21 days	↓CSCs, ↓spheroid-forming ability, ↓survival, migration, and transformation of the HNSCC cells	Upadhyay et al. [112]
Liver	Huh7, Huh6, HepG2, Hep3B, PLC/PRF/5, SKHep1, HLE, and THLE-5b cells	L-685,485 DAPT	10 μmol/L	24–168 h	↓cell growth EpCAM^+^ cells, ↓CSCs, ↓HES1	Kawaguchi et al. [113]
Liver	Hep3B, Huh7, PLC, MHCC97L, and MHCC97H cells Liver cancer spheres	PF-03084014	0.25–2 μM	1–14 days	↓CSCs, ↓proliferation, ↓self-renewal ability	Wu et al. [114]
Liver	MHCC97H, PLC/PRF/5, and HepG2 cells	MRK003	10 μM	1–7 days	↓CSCs, ↓sphere formation ability	Cao et al. [115]
Liver	MHCC97H and MHCC97L cells	PF-03084014 PF-03084014 + sorafenib	0.1–0.25 μM	24 h	↓CSCs, ↓self-renewal ability, ↓proliferation, ↓spheroid formation	Yang et al. [116]
Lung	NSCLC tumor-propagating Cells	DAPT	100 μm	1–2 weeks	↓self-renewal, ↓ tumor propagation, ↓Hes1 and Hey1, ↓CSCs	Zheng et al. [119]
Lung	A549, H460, PC9, H1299, and H661 cells	DAPT DAPT + Cisplatin	10 μmol/L	30 min–48 h	↓CD133^+^ cells, ↑sensitivity to doxorubicin and paclitaxel, ↓Hes-1	Liu et al. [120]
Lung	A549 human lung adenocarcinoma cell line	DAPT	25–75 μM	48 h	↓LCSCs, ↓CD44^+^/CD24^−^ cells	Cai et al. [121]
Lung	NSCLC and SCLC cells	DAPT	25 μmol/L	3–14 days	↓ALDH^+^ cancer cells, ↓proliferation, ↓clonogenicity, ↑cell-cycle arrest, ↓CSCs	Sullivan et al. [122]
Lung	A549 cell line	DAPT CDDP + DAPT	2 μM	48 h	↑cell-cycle arrest, ↓CSCs, ↓proliferation of CD133^+^ and CD133^−^ cells	Liu et al. [123]
Lung	LCSCs and NSCLC cells	RO4929097	1–10 µM	24–48 h	↓p-STAT3,↓self-renewal, ↓LCSCs, ↓HES1	Zhang et al. [124]
Lung	H1299, H441, H460, H358, and A549 cells	MRK-003 and MRK-003 + Docetaxel	5–20 μM	24–48 h	↓CSCs, ↓sphere formation ability, ↓self-renewal, ↓NICD2	Hassan et al. [125]
Lung	HCC2429, HCC827, H358, and HCC4006 cells	PF-03084014 PF-03084014 + erlotinib	1 μM	7 days	↓CSCs, ↓ALDH^+^ cells, ↓EGFR	Arasada et al. [126]
Lung	H23, A549, H358, H661, H1437, H1299, H1703, H520, and ChagoK1 cells	DBZ DBZ + ANF	5 μM	3–30 days	↓oncosphere growth, ↓cell viability, ↓soft agar growth	Ali et al. [127]
Lung	LLC cells	DFPAA	-	7 days	↓NG2^+^ cells, ↓CSCs	Patenaude et al. [128]
Melanoma	B16F10 cells	DFPAA	-	7 days	↓NG2^+^ cells, ↓CSCs	Patenaude et al. [128]
Melanoma	B16F10 and B16F1, SK-MEL-28, A375, and SK-MEL-2 cell lines	DAPT and L-685,458	5–15 μM	12 h–4 weeks	↓CSCs, ↓CD133, ↓metastasis, ↓melanoma growth, ↓angiogenesis, ↓CD133-dependent MAPK signaling	Kumar et al. [129]
Melanoma	WM852c, SK-MEL 28, 1205Lu, HT144, A375, and 451Lu cells	GSI-I GSI-I + ABT-737	0.83 μM	24–48 h	↓primary sphere formation, ↓ALDH^+^ cells, ↓cell viability, ↑apoptosis of the non-MICs, ↓self-renewability	Mukherjee et al. [130]
Melanoma	B16F10, A375, A875, MUM-2C, and MUM-2B cell lines Tumorospheres and the multicellular tumor spheroid (MTS) model	DAPT	10 μM	24–72 h	↓CSCs, ↑E-cadherin, ↓VE-cadherin and Twist1, ↓metastasis	Lin et al. [131]
Osteosarcoma	U2OS, 143B, and MG63 cell lines	DAPT RO4929097	20 μM	24 h	↓spheroid-forming ability, ↓CSCs	Yu et al. [132]
Osteosarcoma	143B, U2OS, and MG-63 cell lines	DAPT DAPT + Cisplatin	5–50 μM	24–72 h	↓OSCs, ↓proliferation, ↓motility, ↑apoptosis, ↑cell-cycle arrest, ↑platinum-sensitivity, ↓ERK and AKT	Dai et al. [133]
Ovarian	SKOV3, A224, OVCAR-3, and UCI-107 cell lines	DAPT	10–20 μg	8 days	↓Colony-formation, ↓SP cells	Vathipadiekal et al. [134]
Ovarian	4412, 4306, OVCAR5, PA-1, OVCAR3, IGROV1, A2780, SKOV3, and OV2008 cells	GSI-I GSI-I + platinum	1–10 μM	1–3 days	↑tumor platinum-sensitivity, ↓CSCs, ↑cell-cycle arrest, apoptosis, and DNA-damage	McAuliffe et al. [135]
Ovarian	SKOV3 and HO8910 cell lines	DAPT	1–20 μg/mL	1–3 days	↓self-renewal ability, ↓proliferation, ↓CSCs, ↓OCSCs-specific surface markers expression, ↓Sox2 and Oct4	Jiang et al. [136]
Ovarian	OVCAR3 and A2780 cells	RO4929097 RO4929097 + CDDP	10 μM 10 mg/kg	24 h	↓proliferation, ↑apoptosis, ↓CSCs	Li et al. [137]
Pancreatic	Bxpc-3 and Panc-1 cell lines	DAPT	1 and 10 μM	48 h	↓CD133^+^, ↓proliferation, ↓CSCs, ↓chemo-resistance	Kang et al. [138]
Pancreatic	BxPC3, KP3 cells	DAPT	2.5–10 µM	48–96 h	↑apoptosis, ↓CSCs, ↓EMT	Palagani et al. [139]
Pancreatic	Panc-1, BxPC-3, MiaPaCa-2, AsPC-1 cell lines	DAPT DAPT + leptin	20 µM	1–10 days	↓PCSCs, ↓proliferation, ↓leptin-induced CD133^+^ and ALDH^+^ cells, ↓tumorsphere formation	Harbuzariu et al. [140]
Pancreatic	CM cell line	DAPT DAPT + 5-FU	2–80 μg/mL	24–48 h	↓clonogenicity, ↑sensitivity to 5-FU, ↓CSC-enriched spheres	Capodanno et al. [141]
Pancreatic	BxPC3 and HPAC cells	DAPT Gem + DAPT	20 μM	72 h	↓CSCs, ↓CD24^+^CD44^+^ cells, ↓pAKT, Hes1, and β-catenin expression, ↓invasion, ↓migration	Lee et al. [142]
Pancreatic	DCLKHI/AcTubHI cells	DAPTMRK-300	DAPT (10–100 nM) MRK-300 (0.72–5 μM)	3–4 days	↓CSCs, ↓AcTub^HI^ cells, ↓PanIN progression, ↓mPanIN epithelial cells expressing Dclk1	Bailey et al. [143]
Pancreatic	Pancreatic cancer cells	MK-0752 GSI + gemcitabine	8 μM	24–72 h	↓CSCs, ↓tumorsphere formation, ↑apoptosis	Abel et al. [144]
Pancreatic	Pa03C, Pa14C, Pa16C, and Pa29C cells	MRK-003 MRK-003 + GEM	2–5 μM	48 h	↓CSCs, ↓NICD, ↓colony-forming capacity, ↑apoptosis	Mizuma et al. [145]
Prostate	DU145 and TRAMP-C2 cell lines and PCSCs	PF-03084014	0/01–100 μM	6 days	↓PCSCs	Wang et al. [147]
Prostate	VCaP and LnCaP96 cell lines	DAPT	1 nM–400 μM	48–96 h	↓CSCs, ↓NICD1	Carvalho et al. [148]
Prostate	Du145, PC3, Du145R, and PC3R cells	PF-03084014 PF-03084014 + docetaxel	0.1–10 μM	48 h	↓CSCs, ↑apoptosis, ↓epithelial to mesenchymal transition, ↓(cyclin E; EGFR, PI3K/AKT, NF-κB, and NF-κB pathway; BCL-XL, BCL-2)	Cui et al. [149]

Abbreviations: 5-FU, fluorouracil; ACC, adenoid cystic carcinoma; ALDH, aldehyde dehydrogenases; Bcl-2, B-cell lymphoma 2; Bcl-xL, B-cell lymphoma-extra-large; BCSCs, breast cancer stem cells; CCSC, colorectal cancer stem cell; CD44, cluster of differentiation-44; CD44ICD, CD44 intracytoplasmic domain; CDDP, cisplatin; *CHI3L1*, chitinase-3-like protein 1; CSCs, cancer stem cells; Cy, cyclopamine; DLL1, Delta-like ligand 1; DLL3, Delta-like ligand 3; Dclk1, Doublecortin-like kinase 1; DBZ, dibenzazepine; E-cadherin, epithelial cadherin; EGFR, epidermal growth factor receptor; EMT, epithelial–mesenchymal transition; EpCAM, epithelial cell adhesion molecule; ER-α, estrogen receptor α; ErbB, epidermal growth factor receptors; ERK, extracellular-regulated kinase; GEM, gemcitabine; GLAST, glutamate aspartate transporter; GSCs, glioma stem cells; GSNO, S-nitrosoglutathione; HIF1α, hypoxia-inducible factor 1α; HNSCC, head and neck squamous cell carcinoma; IL-6, interleukin 6; IL-8, interleukin-8; ILK, integrin-linked kinase; KLF4, Kruppel-like factor 4; LCSCs, lung cancer stem cells; Lgr-5, leucine-rich repeat-containing G-protein coupled receptor 5; MAPK, mitogen-activated protein kinase; Mist1, Muscle, intestine, and stomach expression 1; mPanIN, murine pancreatic intraepithelial neoplasia; MSFE, mammosphere-forming efficiency; NF-κB, nuclear factor-κB; NICD, Notch intracellular domain; N1-ICD, Notch1 intracellular domain; Oct4, octamer-binding transcription factor 4; OSCs, osteosarcoma stem cells; PanIN, pancreatic intraepithelial neoplasia; PCSCs, prostate cancer stem cells; PI3K, phosphatidylinositol-3-kinase; p-STAT3, phosphorylated-signal transducer and activator of transcription 3; Qu, quercetin; RT, radiation; SCUBE2, signal peptide CUB EGF-like domain-containing protein 2; SKP2, S-phase kinase-associated protein 2; SOX2, SRY-box transcription factor 2; SP cells, side population cells; STAT3, signal transducer and activator of transcription 3; TIC, tumor initiating cell; TMZ, temozolomide; TNF-α, tumor necrosis factor-α; TPA, tetradecanoyl phorbol acetate; Twist1, Twist family BHLH transcription factor 1; VE-cadherin, vascular endothelial-cadherin.

**Table 2 molecules-26-00972-t002:** Potential anti-CSC effects and related mechanisms of action of GSIs based on in vivo studies.

Cancer Type	Animal Model	Type and Properties of GSIs	Dose	Exposure time	Major Outcome	References
Adenoid cystic carcinoma	Athymic NCr-nu/nu mice bearing Accx11 cells	DAPT DAPT + radiation	50 mg/kg	35 days	↓SKP2 and N1ICD, ↓growth of ACC, ↓CD133^+^ cells, ↑p27Kip, ↑radio-sensitivity	Panaccione et al. [33]
Blood (leukemia)	Tal1/Lmo2 transgenic mice bearing Tal1/Lmo2 leukemic cells	MRK-003	150 mg/kg	1–3 weeks	↓leukemia-initiating cells	Tatarek et al. [37]
Brain (Glioma)	Immunocompromised mice bearing GBM neurosphere cells	DAPT	10 µM	7 days	↓brain CSCs	Kristoffersen et al. [47]
Brain (medulloblastoma)	BALB/c nude mice xenograft models	DAPT DAPT + HBMEC/D341Med		40 days	↓tumor size and volume	Wang et al. [50]
Brain (Glioma)	Balb/c mice bearing GBM stem cells	DAPT GSI-loaded MLs DAPT + INCB3619	0.5 mg/mL	3 weeks	↑survival rate, ↓CSCs	Floyd et al. [52]
Brain (Glioma)	Nude (nu/nu) mice bearing GICT25 cells	DAPT BMS-708163 RO4929097 RO4929097 + BMS-708163	10 mg/kg (RO4929097) 20 mg/kg (BKM120)	5 weeks	↓tumor growth, ↑survival rate	Saito et al. [54]
Brain (Glioma)	Flank and Intracranial Xenograft tumors nude mice bearing GBM neurosphere cells	GSI-18 MRK-003	2 μM (GSI-18) 2–10 μM (MRK-003)	6 weeks	↓tumor growth, ↑survival rate	Fan et al. [56]
Brain (Medulloblastoma)	Athymic (nude) mice tumor xenografts	GSI-18	0/5 mg	5 days	↓clonogenicity, ↓tumor growth	Fan et al. [57]
Brain (Glioma)	Subcutaneous Xenografts Athymic Nude Mice bearing U251 cells	GSI-I with radiaton	2.5 mmol/L	2–4 weeks	↓tumor growth	Lin et al. [63]
Brain (Glioma)	Immunocompromised mice bearing U87-MG cells	GSI-I GSI-I + TMZ/cyclopamine	−	−	↓CD133^+^ cells, ↓CSCs, ↑TMZ therapeutic effect	Ulasov et al. [64]
Brain (Glioma)	T4302, T4105, and T4597 xenograft tumors Athymic nude mice bearing T4105 CD133^+^ cells	RO4929097 RO4929097 + Farnesyltransferase	30 mg/kg	5–20 days	↑radio-sensitizing, ↓tumor growth	Ma et al. [66]
Breast	Athymic nude nu/nu mice bearing HCC1806 cells	DAPT	10–40 µM	7–21 days	↓tumor formation	Li et al. [73]
Breast	Athymic nude nu/nu mice bearing 231-BR cells	DAPT	8 mg/kg	14 days	↓micro- and macro-metastases	McGowan et al. [75]
Breast	Nude mice bearing MDA231BoM cells	DAPT	−	−	↓invasion	Xing et al. [76]
Breast	Nude mice bearing MDA-MB-231 1833 cells	DAPT RT + DAPT	10 mg/kg	0–20 days	↓tumor growth	Boelens et al. [77]
Breast	NSG mice bearing breast CSCs	DAPT DAPT + Gefitinib			↓Oestrogen effect	Harrison et al. [79]
Breast	Chicken eggs	DAPT DAPT loaded glucose-functionalized MSNs	5 μg/mL	5 days	↓number of cancer cells per mg/tissue	Mamaeva et al. [82]
Breast	NOD/SCID mice bearing Sum159 and MCF7 cells	MRK-003 MRK-003 + docetaxel	75 mg/kg	3–10 weeks	↓breast CSCs, ↓self-renewal, ↓tumor initiation ability	D’Angelo et al. [87]
Breast	FVB/N mice bearing mammospheres	MRK-003	150 mg/kg	2 weeks	↓viability, ↓reduced tumor-resident TIC	Kondratyev et al. [88]
Breast	NOD/SCID mouse bearing MDA-MB231, MDA-MB231BrM, CN34, and CN34BrM cells	Compound E	10 mg/kg	4 weeks	↓growth, ↓metastasis	Xing et al. [89]
Breast	Athymic nude mice bearing MCF7, MDA-MB-231, and BT474 cells	DBZ	1 mg/mL	18–28 days	↓tumor size and volume, ↑mice latency	Harrison et al. [91]
Breast	Nude mice bearing SUM149 cells	MK-0752	25 μM and 25 mg/kg	10 weeks	↓tumor growth	Wang et al. [93]
Breast	SCID/Beige mice bearing human tumorgrafts	MK-0752 MK-0752 + Docetaxel	100 mg/kg	3–21 days	↓primary and secondary MSFE, ↓ALDH^+^ and CD44^+^/CD24^−^ subpopulations, ↓NICD, ↓Hes1, Hey1, Hes5, and myc, ↓tumor growth, ↓BCSCs	Schott et al. [31]
Breast	Balb/C nude mice bearing MDA-MB-231-luc cells	RO4920927	30 mg/kg/day	2 weeks	↓tumor growth	Azzam et al. [94]
Cholangiocarcinoma	(NOD/SCID) female mice bearing HuCCT1, TFK-1 and RBE cell lines	DAPT DAPT + GEM	40 μM	10 days	↓mice tumorigenicity, ↓viability	Aoki et al. [118]
Colorectal	Lgr5-EGFP-creER^T2^ transgenic mouse bearing colorectal cancer cells	DAPT	−	−	↓Lgr5-GFP+ ISC, ↓Ascl2 levels, ↓CSCs	Bu et al. [97]
Colorectal	athymic (nu^+^/nu^+^) mice bearing CRC cells	PF-03084014 PF-03084014 + irinotecan	125 mg/kg	28 days	↓tumor recurrence, ↓tumor growth, ↓ALDH^+^ population, ↓CSCs	Arcaroli et al. [102]
Gastric	Mist1- CreERT2;R26-mTmG mice bearing gastric cancer cells	DBZ	30 μmol/kg	14 days	↓corpus organoid growth	Hayakawa et al. [103]
Gastric	Nude mice bearing MKN45 cells	DAPT	10 mg/kg/body weight	1 week	↓migration, ↓invasion	Barat et al. [104]
Gastric	Nude mice bearing MKN-45 cells	DAPT	10 mg/kg/body weight	5 weeks	↓tumor growth, ↓invasion	Li et al. [105]
Head and neck	Balb/c Nude mice bearing, SAS, OECM1, and FADU cells	DAPT DAPT + Cetuximab	100 mg/kg	6 weeks	↓viability, ↓tumor growth	Chen et al. [108]
Head and neck	Immunocompro-mised mice bearing esophageal adenocarcinoma cells	DAPT DAPT + 5-FU	20 mg/kg	2–10 weeks	↓tumor growth↑sensitivity to chemotherapeutic agents	Wang et al. [109]
Head and neck	BALB/c nude mice bearing CAL27 or SCC9 cells	DAPT DAPT + chemotherapeutic agents	10–20 mg/kg	2 weeks	↓tumor self-renewal capacity	Zhao et al. [111]
Liver	NOD-SCID mouse bearing Huh7 cells	L-685,485 DAPT	5 mg/kg (L-685,458) 20 mg/kg (DAPT)	2,6 weeks	↓tumor growth	Kawaguchi et al. [113]
Liver	Nude or SCID-beige mice bearing MHCC97H and MHCC97L cells	PF-03084014	100 mg/kg	4 weeks	↓tumor metastasis	Wu et al. [114]
Liver	SCID mice bearing 97H spheroid-derived cancer cells	PF-03084014 PF-03084014 + sorafenib	100 mg/kg/day	2–4 weeks	↓spheroid formation	Yang et al. [116]
Lung	Nude mice bearing H460 cells	DAPT DAPT + Cisplatin	2 mg/kg	4 days	↑sensitivity to doxorubicin and paclitaxel	Liu et al. [120]
Lung	NSG mice bearing LCSCs and NSCLC cells	RO4929097	5 mg/kg	10 days	↑platinum sensitivity	Zhang et al. [124]
Lung	NOD/SCID mice bearing H1299 cells	MRK-003 and MRK-003 + Docetaxel	-	3 weeks	↓tumorigenicity	Hassan et al. [125]
Lung	Immunodeficient mice bearing A549 cells	DBZ DBZ + ANF	200 μg/kg	8 weeks	↓tumor growth	Ali et al. [127]
Melanoma	NCRNU nude mice bearing HT144 and WM852c cells	GSI-I GSI-I + ABT-737	0.83 μM	21 days	↓tumor-initiating capacity	Mukherjee et al. [130]
Osteosarcoma	NOD/SCID mice bearing 143B cells	DAPT	10 mg/kg/d	2 weeks	↓tumor recurrence	Yu et al. [132]
Osteosarcoma	BALB/c-nu/nu nude mice bearing 143B cells	DAPT DAPT + Cisplatin	8–10 mg/kg/d	5 weeks	↑platinum-sensitivity, ↓metastasis, ↓tumor growth	Dai et al. [133]
Ovarian	Balb/C athymic mice bearing SKOV3-SP and MP cells	DAPT	5 mg/mL	8 weeks	↓Colony-formation, ↓SP cells	Vathipadiekal et al. [134]
Ovarian	SCID mouse bearing PA-1/luc, OVCAR5/luc, and SKOV3/luc cells Tumor xenografts	GSI-I GSI-I + platinum	5 mg/kg	18 days	↑tumor platinum-sensitivity	McAuliffe et al. [135]
Ovarian	BALB/c nude mice bearing OVCAR3 and A2780 cells	RO4929097 RO4929097 + CDDP	10 mg/kg	6 weeks	↓tumor growth, ↓tumor volume	Li et al. [137]
Pancreatic	Nude mice bearing BxPC3, KP3 cells	DAPT	10 mg/kg/body weight	5 weeks	↓tumorigenesis	Palagani et al. [139]
Pancreatic	Chicken eggs	DAPT DAPT + 5-FU	2–80 μg/mL	11 days	↓tumourigenicity, ↑sensitivity to 5-FU	Capodanno et al. [141]
Pancreatic	KC^Pdx1^, KPC^Pdx1^, and KC^iMist1^ mice bearing pancreatic cancer cells	MRK-300	100 mg/kg	11–13 weeks	↓tumorigenesis, ↓Dclk1-expressing cells	Bailey et al. [143]
Pancreatic	NOD/SCID mice bearing pancreatic cancer cells	RO4929097 GSI + gemcitabine	30 mg/kg	5 days	↓tumor growth	Abel et al. [144]
Pancreatic	athymic nude mice bearing Pa03C, Pa14C, Pa16C, and Pa29C cells	MRK-003 MRK-003 + GEM	150 mg/kg	3 weeks	↓tumor cell proliferation, ↑intratumoral necrosis	Mizuma et al. [145]
Prostate	nu/nu athymic mice bearing Panc215, Panc266, Panc354, and Panc265 xenografts	PF-03084014 PF-03084014 + GEM	150 mg/kg	4 weeks	↓NICD, ↓Hes-1 and Hey-1, ↓CSCs, ↓angiogenesis, ↓proliferation, ↓tumor growth, ↓metastasis, ↑apoptosis, ↓angiogenesis, ↑tumor regression	Yabuuchi et al. [146]
Prostate	(NOD/SCID) mice bearing Du145, Du145R, PC3, and PC3R cells	PF-03084014 PF-03084014 + docetaxel	150 mg/kg	4 weeks	↓tumor growth	Cui et al. [149]

Abbreviations: 5-FU, fluorouracil; ACC, adenoid cystic carcinoma; ALDH, aldehyde dehydrogenases; Ascl2 levels, achaetescute-like 2; Dclk1, Doublecortin-like kinase 1; DBZ, dibenzazepine; E-cadherin, epithelial cadherin; GEM, gemcitabine; ISC, intestinal/colon stem cell; KLF4, Kruppel-like factor 4; LCSCs, lung cancer stem cells; Lgr-5, leucine-rich repeat-containing G-protein coupled receptor 5; Mist1, Muscle, intestine and stomach expression 1; NICD, Notch intracellular domain; SKP2, S-phase kinase-associated protein 2; TIC, tumor initiating cell; TMZ, temozolomide.

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
