# Peer review of "Unlocking the Secrets of Cancer Stem Cells with γ-Secretase Inhibitors: A Novel Anticancer Strategy"

_molecules, 2021, doi:10.3390/molecules26040972_

Round 1

Reviewer 1 Report

General Comments:

This article is concerned with the studies on the effect of γ-secretase inhibitors (GSIs) on various cancer stem cells (CSCs) via modulation of the Notch signaling pathway. It is a collection of studies for the use of GSIs in various cancers, which represents an important area of anti-cancer therapeutics. However, the readability and significance of the review is hindered by the sequential summation of each study without sufficient critical or comparative discussion, as such it reads as a compendium of studies.  

The authors conclude that clinical trials are warranted to assess the use of GSIs in treatment of human cancers (line 656-657), however, current GSI clinical trials are not discussed. In addition, the description of the overall effect of GSIs on CSCs could be improved if the authors described and compared the findings of GSIs on cancer cells and CSCs (lines 642-644). Furthermore, the authors state that the side effects of GSIs may be an impediment to their use (lines 651-652), but this is also not discussed/described in the review.

This manuscript could be improved by inclusion of comparative summation of the studies complied, particularly in Section 5. Furthermore, presentation of Table 1 at the end of section 5 may be more helpful than at the beginning. Additionally, Figure 2 does not supplement the text, perhaps further discussion of the importance of chemical structure of the GSIs should be included.

Specific comments:

Abstract, Line 32: “Despite the feasibility of GSIs for CSC-targeting therapy, the majority of current studies are at basic pharmacological research or preclinical levels, thus, clinical trials are warranted to assess their advantages in the management and control of human cancers.” This statement is not represented in the review as written.

Introduction, Lines 52-60: Additional information on the characterization of CSCs should be included here or in section 5 when discussing various cancer types.

Introduction, Lines 58-60: “Moreover, sphere formation assay…” Incomplete sentence, please rephrase.

Figure 1 Legend, Line 87: Delete the following sentence “This is a figure. Schemes follow the same formatting.”

Figure 1: May prove more informative in Section 2 when discussing Notch signaling pathway.

Section 2 and 3 (Lines 99 and 126):  Headings are not informative and should be clarified.

Author Response

The authors of this manuscript express their sincere thanks to the reviewer for the critical assessment of this work. The authors have acted upon the recommendations of the reviewer which have resulted in a significant enhancement in the quality of this manuscript. All modifications incorporated in the manuscript are highlighted in red color font. A “point-by-point” response to each and every comment is outlined below.

General Comments:

Comment 1:

This article is concerned with the studies on the effect of γ-secretase inhibitors (GSIs) on various cancer stem cells (CSCs) via modulation of the Notch signaling pathway. It is a collection of studies for the use of GSIs in various cancers, which represents an important area of anti-cancer therapeutics. However, the readability and significance of the review is hindered by the sequential summation of each study without sufficient critical or comparative discussion, as such it reads as a compendium of studies. 

Response:

We thank the reviewer for their expertise, time, and effort in reviewing our manuscript. As recommended by the reviewer, we have added a new section to provide a critical discussion of various studies (Section 6, page 28, lines 673-700).

Comment 2:

The authors conclude that clinical trials are warranted to assess the use of GSIs in treatment of human cancers (line 656-657), however, current GSI clinical trials are not discussed. In addition, the description of the overall effect of GSIs on CSCs could be improved if the authors described and compared the findings of GSIs on cancer cells and CSCs (lines 642-644). Furthermore, the authors state that the side effects of GSIs may be an impediment to their use (lines 651-652), but this is also not discussed/described in the review.

Response:

The reviewer has raised an important point here. The statement about clinical trials and the use of GSIs in the treatment of human cancers has been modified (page 29, lines 722-730).

We appreciate the valuable comments about comparing the anticancer effect of GSIs and CSCs. Additional information about GSIs on cancer cells and CSCs has been added to the manuscript (page 29, lines 712-714).

We are grateful for the reviewer’s close observation. Additional information about the side effects of GSIs has been added to the manuscript (page 2, lines 79-82).

Comment 3:

This manuscript could be improved by inclusion of comparative summation of the studies complied, particularly in Section 5. Furthermore, presentation of Table 1 at the end of section 5 may be more helpful than at the beginning. Additionally, Figure 2 does not supplement the text, perhaps further discussion of the importance of chemical structure of the GSIs should be included.

Response:

We greatly appreciate this though-provoking comments. As recommended by the reviewer, we have provided a critical discussion of various studies in the manuscript (Section 6, page 28, lines 673-700).

In order to optimize presentation, Table 1 has been divided into two tables (Table 1 for in vitro studies and Table 2 for in vivo studies). These tables have been moved to the end of section 5 (pages 17-28).

Also, information about GSIs and the importance of their chemical structures has been added to the manuscript (page 5, lines 144-155 and lines 167-171).

Specific comments:

 Comment 1:

Abstract, Line 32: “Despite the feasibility of GSIs for CSC-targeting therapy, the majority of current studies are at basic pharmacological research or preclinical levels, thus, clinical trials are warranted to assess their advantages in the management and control of human cancers.” This statement is not represented in the review as written.

Response:

Thank you for your valuable comment. The statement about clinical trials and the use of GSIs in the treatment of human cancers has been modified (page 1, lines 32-34).

Comment 2:

Introduction, Lines 52-60: Additional information on the characterization of CSCs should be included here or in section 5 when discussing various cancer types.

Response:

We greatly appreciate this comment. Additional information on the CSCs characterization has been added to the manuscript (page 2, lines 54-62).

Comment 3:

Introduction, Lines 58-60: “Moreover, sphere formation assay…” Incomplete sentence, please rephrase.

Response:

The reviewer has raised an important point here. The sentence has been rephrased (page 2, lines 59-62).

Comment 4:

Figure 1 Legend, Line 87: Delete the following sentence “This is a figure. Schemes follow the same formatting.”

Response:

We agree with this and have deleted the sentence.

Comment 5:

Figure 1: May prove more informative in Section 2 when discussing Notch signaling pathway.

Response:

We do not fully comprehend what the review has suggested. The role of the Notch signaling pathway in cancer has been presented in Section 2. We have moved Figure 1 after this section. We have provided explanatory text in the legend of Figure 1.

Comment 6:

Section 2 and 3 (Lines 99 and 126):  Headings are not informative and should be clarified.

Response:

We value the reviewer’s opinion. Accordingly, we have modified the section headings (page 2, line 90 and page 4, line 130).

Additionally,

  1. The reference list has been modified as we have added several new references. Special attention is given to conform to the order of references and bibliographic style of the journal.
  2. The entire manuscript has been thoroughly checked and edited to ensure uniform style, organization, and quality.

On behalf of my co-authors, I once again express my sincere thanks to the erudite reviewers for the valuable suggestions and constructive input to improve the quality of our manuscript.

Reviewer 2 Report

The authors collected detailed information about different studies exploring the effects of GSIs on different types of tumor models and CSCs in vitro an in vivo. The review is well written and I do not have major critics on this review. Readers, interested in Notch signaling, GSIs, CSCs and cancer can use this collection of studies as a helpful introduction to go deeper and more specific into the literature.

minor points:

The very detailed table 1 could be improved. The ordering of the studies (type of malignancy should also be given in the table. I recommend providing this table as an excel-file in a supplement. The interested reader then could perform his own ordering different from the cancer type as well as specific searches. In addition, by using a supplemental excel file, a short comment of the authors regarding study design models (cells, mice, patients) and outcome could be added. If in a specific study in vitro and in vivo experiment were performed in parallel (for example Liu et al., 2013 or Zhang et al., 2017), a subdivision of the in vitro and in vivo information would also be helpful.

In my view the conclusions section is very short. Given the massive side effects of GSIs (for example DAPT) the authors should also mention alternative strategies to target Notch signaling. There are small molecules reported (CB-103, RIN1...) which might target the DNA binding of CSL or the NICD/CSL/MAML interaction to down-regulate Notch signaling. These novel approaches should be added by the authors, at least in the conclusions/outlook section.

Author Response

The authors of this manuscript express their sincere thanks to the reviewer for the critical assessment of this work. The authors have acted upon the recommendations of the reviewer which have resulted in a significant enhancement in the quality of this manuscript. All modifications incorporated in the manuscript are highlighted in red color font. A “point-by-point” response to each and every comment is outlined below.

General comments:

The authors collected detailed information about different studies exploring the effects of GSIs on different types of tumor models and CSCs in vitro an in vivo. The review is well written and I do not have major critics on this review. Readers, interested in Notch signaling, GSIs, CSCs and cancer can use this collection of studies as a helpful introduction to go deeper and more specific into the literature.

Response:

We thank the reviewer for their expertise, time, and effort for reviewing our manuscript. We are deeply encouraged by the generous comments about the quality of our work.

Minor points:

Comment 1:

The very detailed table 1 could be improved. The ordering of the studies (type of malignancy should also be given in the table. I recommend providing this table as an excel-file in a supplement. The interested reader then could perform his own ordering different from the cancer type as well as specific searches. In addition, by using a supplemental excel file, a short comment of the authors regarding study design models (cells, mice, patients) and outcome could be added. If in a specific study in vitro and in vivo experiment were performed in parallel (for example Liu et al., 2013 or Zhang et al., 2017), a subdivision of the in vitro and in vivo information would also be helpful.

Response:

We greatly appreciate this though-provoking comment. Table 1 has been divided into 2 tables: Table 1 for in vitro studies and Table 2 for in vivo studies. Additionally, the studies have been reordered alphabetically based on the type of malignancy.

Comment 2:

In my view the conclusions section is very short. Given the massive side effects of GSIs (for example DAPT) the authors should also mention alternative strategies to target Notch signaling. There are small molecules reported (CB-103, RIN1...) which might target the DNA binding of CSL or the NICD/CSL/MAML interaction to down-regulate Notch signaling. These novel approaches should be added by the authors, at least in the conclusions/outlook section.

Response:

We agree with this suggestion and have added various novel approaches to overcome the challenges related to the side effects of GSIs to the conclusions section (page 29, lines 722-730) as recommended by the reviewer.

Additionally,

  1. The reference list has been modified as we have added several new references. Special attention is given to conform to the order of references and bibliographic style of the journal.
  2. The entire manuscript has been thoroughly checked and edited to ensure uniform style, organization, and quality.

On behalf of my co-authors, I once again express my sincere thanks to the erudite reviewers for the valuable suggestions and constructive input to improve the quality of our manuscript.

Round 2

Reviewer 1 Report

The authors have adequately addressed the reviewer’s comments and as a result the manuscript has been greatly improved in both readability and discussion of the importance of Notch signaling pathway modulation via GSIs.